# Rotational drift in Antarctic sea ice: pronounced cyclonic features and differences between data products

Wayne de Jager[1], Marcello Vichi[1,2]

[1]Department of Oceanography, University of Cape Town, Cape Town, 7700, South Africa
[2]Marine and Antarctic Research Centre for Innovation and Sustainability (MARIS), University of Cape Town, Cape Town, 7700, South Africa

*Correspondence to*: Wayne de Jager (djgway001@myuct.ac.za)

**Abstract.** Sea-ice extent variability, a measure based on satellite-derived sea ice concentration measurements, has traditionally been used as an indicator to evaluate the impact of climate change on polar regions. However, concentration-based
measurements of ice variability do not allow to discriminate the relative contributions made by thermodynamic and dynamic processes, prompting the need to use sea-ice drift products and develop methods to quantify changes in sea-ice dynamics that would indicate trends in the ice characteristics. Here, we present a new method to automate the detection of rotational drift features in Antarctic sea ice from space at spatial and temporal scales comparable to that of polar weather. This analysis focusses on drift features in the Atlantic Sector of the Southern Ocean in the period 2013–2020 using currently available
satellite ice motion products from EUMETSAT OSI SAF. We observe a large discrepancy between cyclonic and anticyclonic drift features, with cyclonic features typically exhibiting larger drift intensity and spatial variability according to all products. The mean intensity of the 95th percentile of cyclonic features is 1.5–2.0 times larger for cyclonic features than anticyclonic features. The spatial variability of cyclonic features increased with intensity, indicating that the most intense cyclonic features are also the least homogenous. There is good agreement between products in detecting anticyclonic features, however, larger
disagreement is evident for cyclonic features with the merged product showing the most intense 95th percentile threshold and largest spatial variability, likely due to the more extended coverage of valid vorticity points. A timeseries analysis of the 95th percentile shows an abrupt intensification of cyclonic features from 2014–2017, which coincides with the record decline in Antarctic sea-ice extent since winter of 2015. Our results indicate the need for systematic assessments of sea-ice drift products against dedicated observational experiments in the weather-dominated Atlantic sector. Such information will allow to confirm
whether the detected increase in cyclonic vorticity is linked to rapidly changing sea-ice dynamics driven atmospheric changes and establish the measure of rotational sea-ice drift as a potential indicator of weather driven variability in Antarctic sea ice.

## 1 Introduction

The Antarctic continent is surrounded by seasonally varying sea ice. During the austral winter, sea-ice coverage expands zonally when progressing northward into the Southern Ocean, and conversely constricts south towards Antarctica's coastline in the austral summer. Sea ice plays a major role in ocean and atmosphere interactions, primarily acting as a heat, mass and momentum exchange modulator between the surface water and overlying air masses (McPhee et al., 1987; Vihma et al., 2014). Sea-ice coverage is a key component of the Southern Ocean climate system and therefore also the global climate system (Mayewski et al., 2009). Antarctic sea ice is characterized by high temporal and spatial variability, which has shown an increase in the recent years, as well as a major abrupt reduction in the circumpolar ice cover (Parkinson, 2019; Turner et al., 2017). The El Niño Southern Oscillation (ENSO) and the Southern Annular Mode (SAM) have also been shown to influence sea-ice variability (Goosse et al., 2009; Pezza et al., 2012; Thompson, 2002; Yuan, 2004). However, the degree of relative influence of these larger scale atmospheric modes is debated (Schroeter et al., 2017), and it is becoming increasingly argued that sea-ice variability trends – especially in the Atlantic Sector – are primarily driven by local weather events rather than larger scale atmospheric features (Kwok et al., 2017; Matear et al., 2015). It has been shown that intense polar cyclones can continually reshape the underlying marginal ice zone (MIZ, traditionally defined as the region where sea-ice concentration is between 15–80%) and pack ice, as the strong winds induce synoptic scale rotation into the ice pack while carrying warm, moist air along their trajectory (Vichi et al., 2019; Wang et al., 2015). Furthermore, it has been suggested that the unprecedented decrease in ice cover between 2014–2016 was partially the result of intense atmospheric storms injecting large scale momentum into the underlying sea ice, causing the ice to drift northward and melt at the warmer lower latitudes (Turner et al., 2017; Wang et al., 2019b). This phenomenon is likely to grow in influence as extratropical cyclones shift poleward and polar storms intensify (Chang, 2017; Tamarin-Brodsky and Kaspi, 2017). It is therefore necessary to consider the effect that local weather systems have on ice dynamics and to evaluate whether these phenomena influence the overall Antarctic sea-ice drift over time.

Sea-ice concentration (SIC, a measure of the proportion of ice-covered water to total area) has been derived from remote sensing through the passive microwave (PM) brightness temperature of the ocean surface (Turner et al., 2016). Trends in sea-ice extent (SIE, a measure of the area of ocean surface covered by 15 % or greater SIC) are commonly used indicators of sea-ice variability in both the Arctic and Antarctic regions, and subsequently presented to highlight the effects of global warming (Masson-Delmotte et al., 2021). Antarctic SIE trends have historically been relatively constant, however, are recently characterized by pronounced variability with regionally distinct trends, with a record high SIE in 2014 declining to a record low SIE in 2017 (Parkinson, 2019; Turner et al., 2017). While temporal variability in SIC and SIE can be directly computed from PM data, identifying the mechanisms driving this variability is less obvious. Dynamic and thermodynamic processes together alter the Southern Ocean's ice landscape (Stevens and Heil, 2011). However, concentration-only based measurements from space do not allow to discriminate their relative contributions. Ice motion products can therefore be used in conjunction with ice concentration products to help distinguish the dynamical component. These products use a feature-tracking method,

whereby a distinguishable pattern in the ice is identified and followed across a temporal sequence of brightness temperature (or backscatter) maps, and the resultant displacement of this pattern over the time interval can then be estimated. This retrieval method, therefore, relies on the persistence of these distinguishable patterns throughout the time interval. Sea-ice movement

is primarily driven by the exchange of momentum from the overlying atmosphere or ocean surface (Biddle and Swart, 2020; Holland and Kwok, 2012). Waves, ocean tilt, Coriolis forcing, ice inertia and internal ice stressors also play a role (Feltham, 2008). Studies using this feature-tracking drift retrieval method have shown that seasonal patterns and long-term trends in Southern Ocean sea-ice drift have a strong correlation with local winds and that ice drift speeds in the period 1992–2010 have increased by up to 30 % (Holland and Kwok, 2012; Kwok et al., 2017). Satellite derived drift speeds between 1982–2015 were

measured to be ~1.4 % of the geostrophic wind, while Arctic drift speeds was approximately half of that over the same period, suggesting that the thinner, weaker and less compact Antarctic sea ice may be more susceptible to wind forcing (Kwok et al., 2017).

Given the recent availability of motion products, changes in rotational features of Antarctic sea ice have not yet been quantified.

This study proposes a method for the detection and quantification of rotational drift features in Antarctic sea ice at temporal and spatial scales similar to that of local weather events. This is done focusing on the Atlantic Sector that is more directly affected by weather variability (Matear et al., 2015), by computing the sea-ice vorticity using satellite ice drift estimates from the European Organisation for Exploitation of Metalogical Satellites (EUMETSAT) Ocean and Sea Ice Satellite Application Facility (OSI SAF) and quantifying the rotational drift of the sea ice within circular domains. This method proposes a measure

of rotational drift of sea ice which could be used as a potential index with which interannual dynamical trends can be analysed. Several ice motion products are operationally available, all of which undergo shared processing chains but use input data measured by different satellite sensors or remote sensing techniques. While a validation of the vorticity estimates is not included due to the sparsity of *in situ* data, a comparison between the relative performance of four drift products is shown. This study aims to investigate the role of polar weather on driving sea-ice dynamics, and to assess to what extent ice drift

products are able to capture changes over the period of data availability (2013–2020).

## 2 Data

Six different products from the EUMETSAT OSI SAF low resolution sea-ice drift product range (or OSI-405-c) were used in this analysis: the multi-sensor merged, Advanced Microwave Scanning Radiometer 2 (AMSR-2), Advanced Scatterometer (ASCAT), Special Sensor Microwave Imager (SSM/I) and two Special Sensor and Microwave Imager/Sounder (SSMIS)

products. The OSI-405-c processing starts from daily maps of brightness temperature, aggregated from swath data measured from the mentioned sensors, namely: the AMSR-2 on Japan Aerospace Exploration Agency (JAXA) platform GCOM-W1; the SSM/I on the Defence Metalogical Satellite Program (DMSP) platform F15, the SSMIS on DMSP platforms F17 and F18 and the backscatter data from the ASCAT sensor on EUMETSAT platform MetOp-A. A Laplacian filter is applied to these

daily maps to enhance specific ice features in the image and an ice-edge mask (OSI-402-c product) is also applied.
Displacement vectors are then computed from two daily images approximately 48 h apart using a feature-tracking method. The temporal range of these single-sensor products in the Southern Hemisphere are as follows: The AMSR-2 product is available from September 2015 to present; the ASCAT from March 2013 to present; the SSM/I from March 2013 to September 2015; the SSMIS-F17 from September 2015 to April 2018; and the SSMIS-F18 from August 2018 to present. For this analysis, motion vectors derived from the SSM/I and SSMIS instruments are grouped to provide a continuous dataset of measurements since 2013. This group will be analysed as a single product and referred to as the SSMI/S product. The multi-sensor merged product implements a two-step process, firstly by using a weighted average of all valid single-sensor data at a particular grid point – where the weighting of each single-sensor product is inversely proportional to the validated error of that product – and secondly by interpolating surrounding vectors for grid points where no data is available from any single-sensor product. The OSI-405-c product range was used because its spatial coverage spans over the entire Antarctic sea ice landscape with a comparably intermediate temporal resolution of approximately 48 h, although the 62.5 km spatial resolution is coarse. The OSI-405-c processing chain uses the Continuous Maximum Cross Correlation method, which includes an additional processing step on the traditionally used Maximum Cross Correlation method. This additional step reduces the high level of quantization noise that typically hinders displacement vector retrieval over short time periods (Lavergne et al., 2010). Vector displacement uncertainties are also included in all OSI-405-c products from 1st June 2017. Antarctic OSI-405-c motion vectors are mapped onto a NSIDC polar stereographic projection (https://nsidc.org/data/polar-stereo/ps_grids.html) and are available from April 2013 to present at near real-time – making it a good candidate for year-to-year variability analysis and operational use. Further information on the OSI-405-c product processing chain can be found in Lavergne et al. (2010). Due to the limitations of measuring sea-ice drift in melting conditions and during periods of insufficient ice cover (Lavergne, 2016; Sumata et al., 2015), only the months of June-October were considered, and our analysis focused on the Atlantic Sector of the Southern Ocean, spanning the area between 65° W and 50° E.

## 3 Methodology

Sea-ice vorticity features were identified using a detection algorithm applied to the single-sensor products (AMSR-2, ASCAT and SSMI/S family) and the merged product. The first step of this process is the computation of the vorticity field for each $\approx$ 48 h dataset. Here, using a regularly spaced fixed grid, the vorticity is computed at every square pixel $P_{i,j}$ from the drift vector estimates at $P_{i-1,j}$, $P_{i+1,j}$, $P_{i,j-1}$ and $P_{i,j+1}$:

$$\zeta_{i,j} = \frac{\delta v_{i,j}}{\delta x} - \frac{\delta u_{i,j}}{\delta y} \cong \frac{v_{i+1,j} - v_{i-1,j}}{2\Delta x} - \frac{u_{i,j+1} - u_{i,j-1}}{2\Delta y} \tag{1}$$

where $\zeta_{i,j}$ is the relative vorticity of a pixel $P_{i,j}$ using a centred-in-space scheme (units: seconds$^{-1}$), $u$ and $v$ are the zonal and meridional sea-ice velocities respectively (units: metres seconds$^{-1}$), and $\Delta x$ and $\Delta y$ is the length and width of a pixel, which in

our case is a 62.5 km square. Low-quality flagged drift estimates (i.e., flag values 0-19 in the OSI-405-c product) were rejected, while only those flagged with a good quality index (i.e., flag values 20-30) were considered. Rejected and good quality index flags are determined by the quality of the drift estimate retrieval and are made available in the OSI-405-c product. Nominal quality estimates (i.e., flag value of 30) have the lowest retrieval uncertainty and were measured independently of their neighbours, while flag values 20-29 included drift estimates that required a correction or interpolation scheme from neighbouring locations and therefore have a larger uncertainty. Rejection quality flags correspond to locations where no valid drift estimate could be made, and therefore no vorticity values are computed at those grid points or their adjacent neighbours (more information on the rejection and quality index flags can be found in Sect. 4.4 of the product user manual: https://osisaf-hl.met.no/sites/osisaf-hl/files/user_manuals/osisaf_cdop2_ss2_pum_sea-ice-drift-lr_v1p8.pdf). The 20-30 range of flag values was chosen for this analysis because each drift estimate has a corresponding drift uncertainty (as from 1$^{st}$ June 2017), and so while some non-nominal drift vectors may be of degraded quality, this potential noise was quantified and propagated through the vorticity computation.

In the second step of the process, the algorithm generates virtual circular subdomains $D_r$ of radius $r$ centred at every grid point in our vorticity field. Each of these subdomains represent a vorticity feature, which can partially overlap one another in space. We define the feature intensity as the mean of all vorticity estimates contained within its circumference, and the feature variability as the standard deviation of all vorticity estimates contained within its circumference. Therefore, a negative mean intensity feature represents a circular area of sea ice dominated by cyclonic rotation, while a positive mean intensity feature represents an area dominated by anticyclonic rotation. A minimum pixel validity threshold of $T$ is applied to every subdomain $D_r$, ensuring that each classified feature has an adequate number of valid vorticity values within its circumference. Subdomains that fail to meet the minimum pixel validity threshold are ignored, thus reducing the algorithm's susceptibility to classifying small regions of intense vorticity at the ice edge or coastline as features. This process is repeated independently per product with varying $r$ (500, 450 and 400 km) and $T$ (90, 85 and 80 %) parameter values, meaning that all identified features contain 180-220 valid vorticity values within their circumference, depending on the choice of $r$ and $T$. The choice of 500, 450 and 400 km radius features is oriented towards capturing the effect of large-scale synoptic features on the underlying sea ice, which are of the order of 1000 km, and therefore the 62.5 km grid spacing of the OSI-405-c product is fine enough to capture these features. All permutations of varying parameter values radius $r$ and threshold $T$ produced similar results, indicating that the presented results are robust to the choice of free parameters. Results shown in Sect. 4 used $r = 450$ km and $T = 90$ %.

An analysis of the propagation of uncertainties in our feature detection was also done. This follows the methods described in Sect. 2.4 of Dierking et al. (2020). Here, we assume that the position error and uncertainty in the timing of the measurements are negligible, and therefore the uncertainty in vorticity of a square cell in a fixed grid can be estimated by:

$$\sigma_{vort}^2 = \frac{2\sigma_{tr}^2}{L^2 \Delta T^2} \tag{2}$$

where $\sigma_{tr}^2$ is the tracking error of the drift vector (unit: metres), $L$ is the length of the square cell (unit: metres), and $\Delta T$ is the time interval between two brightness temperature swath measurements (unit: seconds). In this case, $L$ remains constant at 62.5 km. For the single-sensor products, $\Delta T$ varies within a 48 h ± 8 h range depending on the timing of the two overlapping satellite swaths, while $\Delta T$ for the merged product is artificially set to 48 h. Therefore, all classified vorticity features have an associated uncertainty, which is defined as the mean of all $\sigma_{vort}$ estimates within the circumference of the feature.

## 4 Results

### 4.1 Case study analysis

Two case studies were considered in this analysis to highlight the usefulness of quantifying synoptic scale rotational features in the sea ice, and to assess the vorticity feature detection algorithm described in Sect. 3. These examples have been selected based on known synoptic events observed during the winter expeditions of the SA Agulhas 2 in the region. Case Study 1 considers an explosive polar storm which formed over the open ocean and propagated south-east over the MIZ and ice interior. Case Study 2 considers a stationary high-pressure cell persisting over the ice interior in the eastern Weddell Sea. These maps illustrate some of the features of the vorticity fields obtained from the available products and the overall functioning of the detection algorithm. An in-depth analysis of the resulting vorticity values and the relative uncertainties is given in Sec. 4.2 and 4.3, in which we will refer to these case studies for comparison.

### 4.1.1 Case Study 1: Explosive polar storm traversing over the sea ice

There existed an explosive atmospheric cyclone moving from the MIZ to over the ice interior between 2–5 July 2017. This cyclone was chosen for a case study as its effects on the underlying sea ice has been well documented with both satellite and *in situ* data (Vichi et al., 2019). Figure 1 shows the vorticity field between midday 3 July and midday 5 July according to each of the four products, with the masking due to rejected drift estimates indicated by the dotted hatching. Overlain mean sea level pressure (MSLP) contours indicate that cyclonic structure in the atmosphere persisted over the ice interior during this 48 h period, with the mean position of its core located at approximately 65° S and 21° E. According to the merged product, the cyclonic vorticity feature in the sea-ice field is consistent with the structure of the atmospheric cyclone over the same period. This is evidenced by the large region of negative vorticity in the sea-ice field (blue) underneath the overlying atmospheric cyclone (Fig. 1d). The magenta ring shown in Fig. 1 represents the location of the most intense cyclonic ice drift feature detected using the merged product, with its 450 km radius centred at 65.7° S and 20.6° E (mean vorticity: -1.13 ×$10^{-3}$ ± 0.90 ×$10^{-6}$ s$^{-1}$). Conversely, none of the single-sensor ice drift products detect this intense feature underneath the atmospheric cyclone (Fig. 1a, 1b and 1c), although they all capture some negative vorticity in the region. In addition to the area of ice beneath the atmospheric cyclone, the wider vorticity field is also associated with the cyclonic and anticyclonic curvature of the isobars. All four products detect positive vorticity beneath the elongated high-pressure ridge over the Weddell Sea,

separating two smaller regions of negative vorticity at the ice edge (approximately 62° S and 30° W) and continental coastline (approximately 77° S and 15° W), each of which lie below a pressure trough. All products have some rejected drift estimates at the MIZ, while the single-sensor products have far more in the ice interior, which results in a patchier vorticity field, especially when it is negative. All four products instead detect a similar vorticity field in the Weddell Sea region, suggesting that good quality drift estimates are more in agreement under anticyclonic drift conditions.

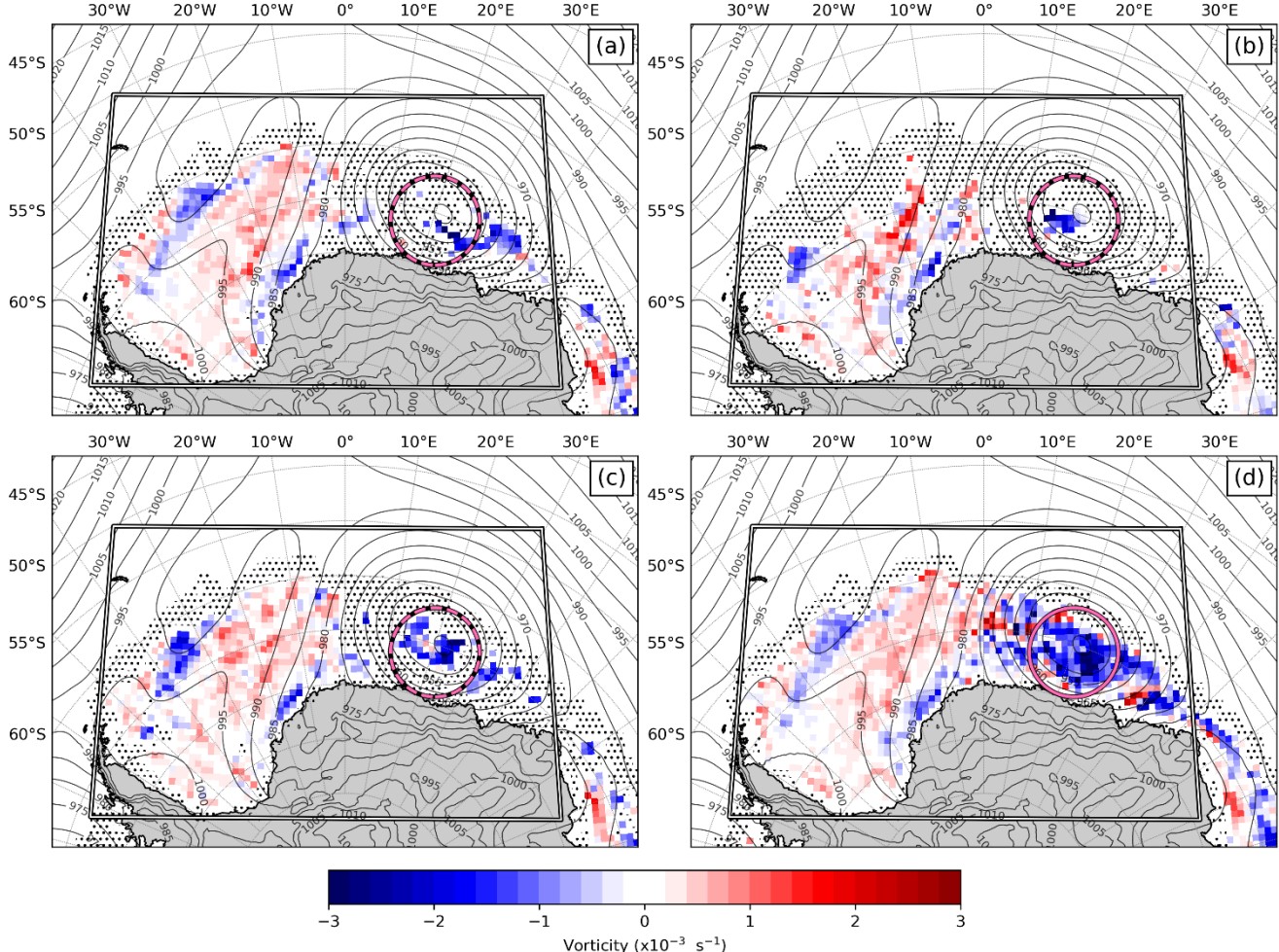

**Figure 1: The sea-ice vorticity field for Case Study 1 between 3 July 2017, 12:00:00 – 5 July 2017, 12:00:00 (UTC) computed using the (a) AMSR-2, (b) ASCAT, (c) SSMI/S and (d) merged ice drift products from EUMETSAT OSI SAF. Overlying contours show the 48 h mean sea level pressure (MSLP) computed from hourly ECMWF-ERA5 reanalysis data over the same period. Ice drift**
**estimates that were not considered in the vorticity computation due to their rejection flag status are shown with dotted hatching, and the rectangular box marks the area boundaries over which the algorithm described in Sect. 3 is applied. The magenta ring shows the location of the most intense cyclonic feature detected by the merged product over this 48 h period, while the dashed rings indicate that the corresponding features did not meet the pixel validity threshold requirement using the AMSR-2, ASCAT and SSMI/S products.**

### 4.1.2 Case Study 2: High-pressure cell persisting over the eastern Weddell Sea

Figure 2 shows the sea-ice vorticity field from approximately midday 21 July to midday 23 July 2019. During this period, ERA5 reanalysis data indicates that a strong high-pressure cell was persistent over the eastern Weddell Sea, with the average position of its core between located at 67.2° S and 18.9° W. The green ring represents the location of the most intense

anticyclonic ice drift feature detected using the merged product, with its 450 km radius centred at 66.5° S and 19.8° W (mean vorticity: $0.63 \times 10^{-3} \pm 0.37 \times 10^{-6}$ s$^{-1}$, Fig. 2d). The passive-microwave based single-sensor products detect similar vorticity fields over the same area, with the AMSR-2 and SSMI/S products measuring $0.63 \times 10^{-3} \pm 0.36 \times 10^{-6}$ s$^{-1}$ (Fig. 2a) and 0.64 $\times 10^{-3} \pm 0.72 \times 10^{-6}$ s$^{-1}$ (Fig. 2c) respectively. The active-microwave based ASCAT product fails to detect this feature due to the insufficient number of valid vorticity points (Fig. 2b). Again, the vorticity fields detected by all four products show a strong

correlation with the curvature of the overlying isobars across the entire rectangular region, much like that shown in Case Study 1 (Sect. 4.1.1). This is particularly visible in Fig. 2 along the negative-to-positive vorticity gradient from the western-to-eastern Weddell Sea, where the parallel isobars lie perpendicular to the vorticity gradient, suggesting the atmospheric pressure gradients control the underlying vorticity field. A region of negative vorticity is also detected near the eastern boundary of the rectangular region which mimics the curvature of the overlying low-pressure cell deflecting eastwards. Similarly to Case Study

1, all three single-sensor products have a higher frequency of rejected drift estimates compared to the merged product (Fig. 2).

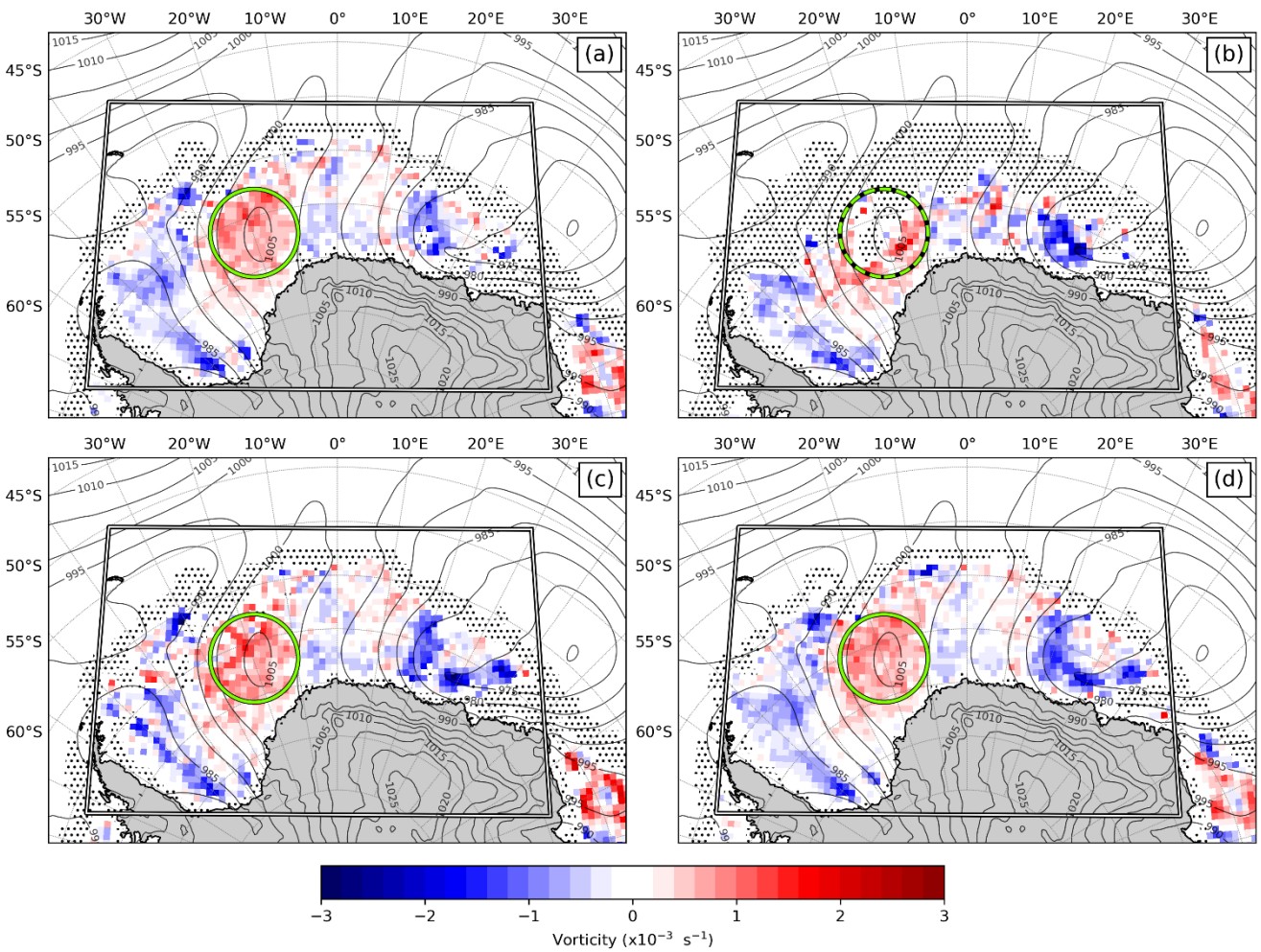

**Figure 2: The sea-ice vorticity field for Case Study 2 between 21 July 2019, 12:00:00 – 23 July 2019, 12:00:00 (UTC) computed using the (a) AMSR-2, (b) ASCAT, (c) SSMI/S and (d) merged ice drift products from EUMETSAT OSI SAF. Overlying contours show the 48 h mean sea level pressure (MSLP) computed from hourly ECMWF-ERA5 reanalysis data over the same period. Ice drift estimates that were not considered in the vorticity computation due to their rejection flag status are shown with dotted hatching, and the rectangular box marks the area boundaries over which the algorithm described in Sect. 3 is applied. The green ring shows the location of the most intense anticyclonic feature detected by the merged product over this 48 h period, while the dashed ring indicates that the corresponding feature did not meet the pixel validity threshold requirement using the ASCAT product.**

## 4.2 Analysis of uncertainties and comparison of detected features in the drift products

Figure 3 shows the distribution of feature intensity relative to its uncertainty for both cyclonic and anticyclonic features. This analysis was done using years 2017–2020 due to the availability of drift uncertainty estimates in the OSI-405-c product. The intensity of rotational features ($\sim 10^{-3}$ s$^{-1}$) is approximately 3 orders of magnitude larger than its associated uncertainty ($\sim 10^{-6}$ s$^{-1}$), mostly due to the large $L^2$ and $\Delta T^2$ terms in the denominator of Eq. (2). The SSMI/S product shows the largest mean

uncertainty for both cyclonic and anticyclonic features, while uncertainties are smallest using the AMSR-2 product (Table 1). Despite differences between the three single-sensor products, each of them show near identical estimates of the mean uncertainty of cyclonic features relative to that of anticyclonic features. Conversely, the merged product shows a larger mean uncertainty for cyclonic features (0.48 $\times10^{-6}$ s$^{-1}$) than anticyclonic features (0.41 $\times10^{-6}$ s$^{-1}$), suggesting that cyclonic drift estimates are noisier than that of anticyclonic ones. Single-sensor derived products show a small range of uncertainties for both cyclonic and anticyclonic features, as indicated by the flatness of scattering in Fig. 3a–f. This is also shown statistically in Table 1, where the standard deviation of the uncertainty spread is approximately 0.04 $\times10^{-6}$ s$^{-1}$ for all three single-sensor products. This is because the tracking error ($\sigma_{tr}^2$) term in Eq. (2) is mostly spatially uniform, although there is some variation introduced into the uncertainty field by the non-nominal quality flagged drift estimates and relatively small deviations in the time interval ($\Delta T$). Conversely, the merged product (Fig. 3g and 3h) shows a far greater standard deviation of uncertainties; 0.15 $\times10^{-6}$ s$^{-1}$ and 0.10 $\times10^{-6}$ s$^{-1}$ for cyclonic and anticyclonic features respectively. This result is intuitive as the merged product is created using a combination of the single-sensor products and its uncertainty is computed using a Gaussian error propagation function based on the uncertainties of its constituents, and thus the resultant uncertainty of the merged product is expectedly more variable than that of the single-sensor products (Lavergne, 2016). The features detected in Case Studies 1 and 2 are shown with magenta and green diamond markers respectively in Fig. 3. The cyclonic vorticity feature detected by the merged product in Case 1 (Fig. 1d) had both an intensity and variability measured beyond the 95th percentiles relative to the other cyclonic features detected from 2017–2020 (dashed lines in Fig. 3h). The anticyclonic feature detected in Case 2 (Fig. 2a, 2c and 2d) is measured beyond the 95th percentile of intensity – but near average uncertainty – relative to the other anticyclonic features detected between 2017–2020, according to the AMSR-2 (Fig. 3a), SSMI/S (Fig. 3e) and merged (Fig. 3g) products.

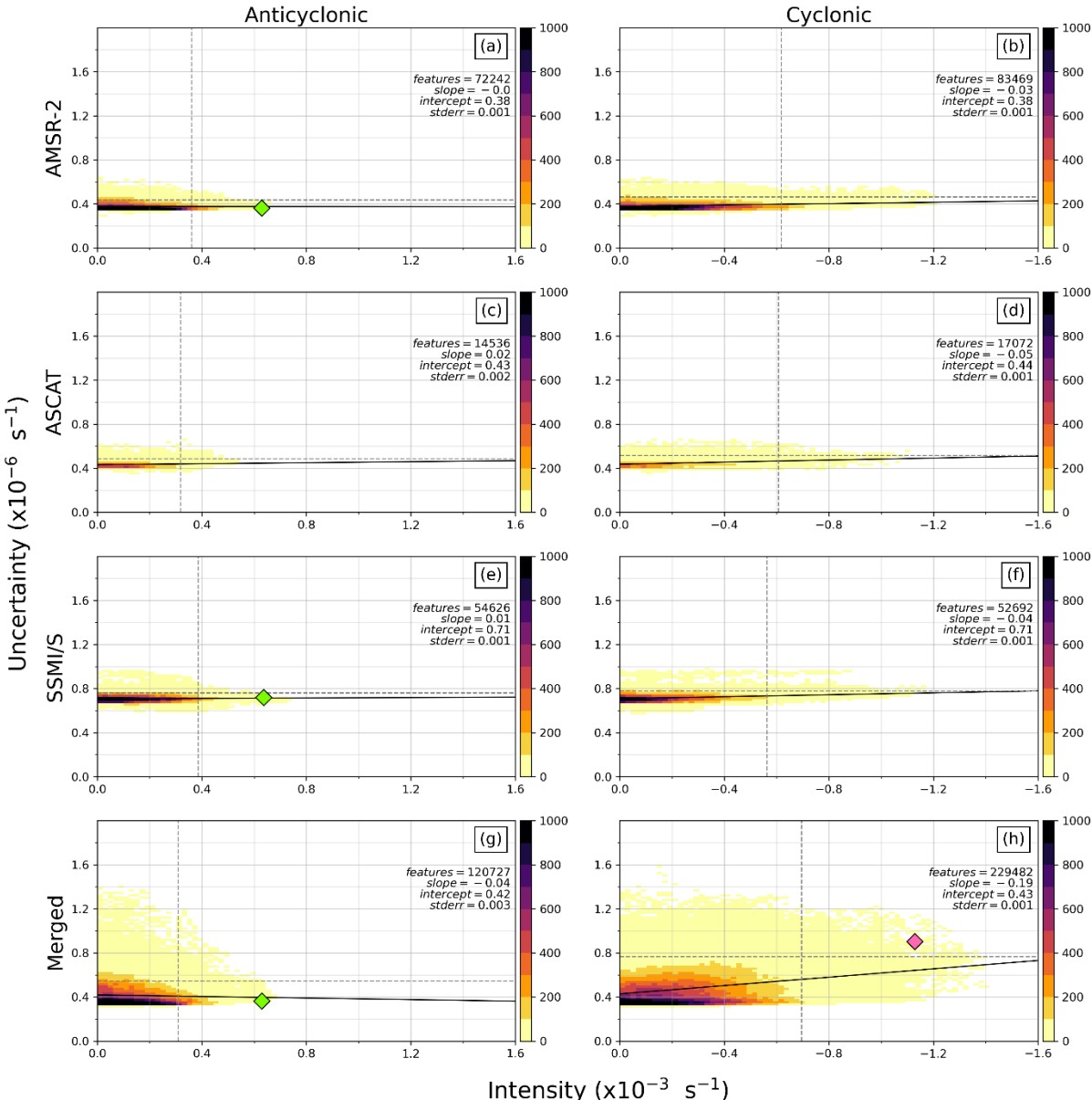

**Figure 3: A two-dimensional histogram of the intensity and uncertainty of vorticity of all features detected between 2017–2020. Anticyclonic and cyclonic features are shown in the left and right columns respectively, while each product is shown in a different row. Panels (a) and (b) show features detected using the AMSR-2 product; panels (c) and (d) using the ASCAT product; panels (e) and (f) using the SSMI/S product; and panels (g) and (h) using the merged product. The lines-of-best-fit are shown with a black solid line, along with its slope, intercept, standard error, and the number of features detected. The vertical and horizontal dashed lines represent the 95th percentiles of the intensity and uncertainty respectively. The colormap indicates the number of features detected in each bin. The anticyclonic feature described in Case Study 2 is shown with a green diamond marker, and the cyclonic feature described in Case Study 1 is shown with a magenta diamond marker (Sect. 4.1). Note the relative scaling order of magnitude between variables, and the reverted x-axis for cyclonic features to aid comparison. The gradients of the lines-of-best-fit reported in the Results (Sect. 4.2) have been computed using the absolute values of the x-axis to simplify the comparison between the cyclonic and anticyclonic distributions.**

**Table 1: The mean and standard deviation of the uncertainty and variability of vorticity of all features detected between 2017–2020 (Format: Mean ± STD). Note the relative scale of units of the uncertainty estimates relative to that of the variability.**

| UNCERTAINTY ($\times 10^{-6}\ s^{-1}$) | AMSR-2 | ASCAT | SSMI/S | MERGED |
|---|---|---|---|---|
| **Anticyclonic** | $0.39 \pm 0.04$ | $0.44 \pm 0.03$ | $0.71 \pm 0.03$ | $0.39 \pm 0.01$ |
| **Cyclonic** | $0.39 \pm 0.03$ | $0.45 \pm 0.04$ | $0.72 \pm 0.04$ | $0.48 \pm 0.15$ |
| VARIABILITY ($\times 10^{-3}\ s^{-1}$) | | | | |
| **Anticyclonic** | $0.33 \pm 0.11$ | $0.38 \pm 0.09$ | $0.38 \pm 0.10$ | $0.39 \pm 0.17$ |
| **Cyclonic** | $0.42 \pm 0.14$ | $0.47 \pm 0.13$ | $0.46 \pm 0.13$ | $0.55 \pm 023$ |

The relationship between the feature intensity and its associated variability within the search radius (Fig. 4) indicated that they are of the same order of magnitude. Here it can be seen that the scale of standard deviation is far more comparable to the feature intensity. It is shown that the mean variability is larger for cyclonic features than anticyclonic features in all four products, of which the largest spread is noted using the merged product (Table 1). Additionally, all four products agree that the variability of cyclonic features is proportional to its intensity – as visualized by the gradient of the black lines-of-best-fit in Fig. 4b, 4d, 4f, and 4h – indicating that the highest intensity cyclonic features are also the most variable. The merged product has the largest slope of 0.44 for cyclonic features (Fig. 4h), and the single-sensor AMSR-2 (Fig. 4b), ASCAT (Fig. 4d) and SSMI/S (Fig. 4f) products have smaller slopes of 0.31, 0.33 and 0.34 respectively. No obvious relationship between intensity and associated variability is seen for anticyclonic features, with the single-sensor AMSR-2 (Fig. 4a), ASCAT (Fig. 4c) and SSMI/S (Fig. 4e) products all showing a slope of approximately 0.01. The merged product has an inversely-proportional slope of -0.10 (Fig. 4g), indicating that intense anticyclonic features are more homogenous. The features detected in Case Studies 1 and 2 are shown in Fig. 4 with magenta and green diamond markers respectively. The cyclonic vorticity feature detected by the merged product in Case 1 lies close to the regression line, with its intensity and variability both measured beyond the 95th percentiles relative to the other cyclonic features detected in the same period (dashed lines in Fig. 4h). This is aligned with the atmospheric event that was reported to be an explosive cyclone (Vichi et al., 2019). Much like the uncertainty distribution shown in Fig. 3, the AMSR-2 (Fig. 4a), SSMI/S (Fig. 4e) and merged (Fig. 4g) products detected the anticyclonic feature described in Case 2 (Sect. 4.1.2) with an intensity measured beyond the 95th percentile but near-average variability relative to the other anticyclonic features.

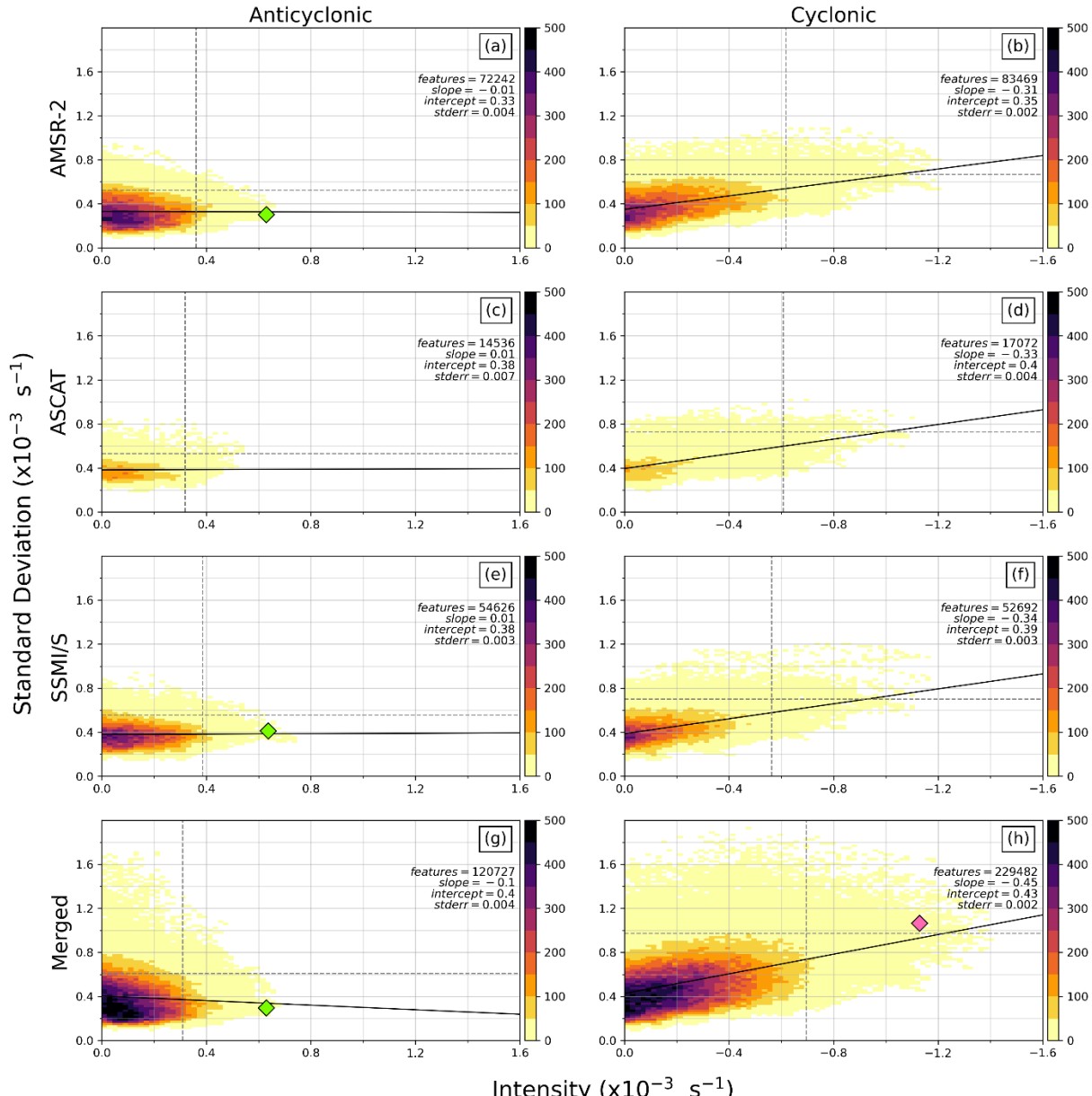

**Figure 4: A two-dimensional histogram of the intensity and standard deviation of vorticity of all features detected between 2017–2020.** Anticyclonic and cyclonic features are shown in the left and right columns respectively, while each product is shown in a different row. Panels (a) and (b) show features detected using the AMSR-2 product; panels (c) and (d) using the ASCAT product; panels (e) and (f) using the SSMI/S product family; and panels (g) and (h) using the merged product. The lines-of-best-fit are shown with a black solid line, along with its slope, intercept, standard error, and the number of features detected. The vertical and horizontal dashed lines represent the 95th percentiles of the intensity and standard deviation respectively. The colormap indicates the number of features detected in each bin. The anticyclonic feature described in Case Study 2 is shown with a green diamond marker, and the cyclonic feature described in Case Study 1 is shown with a magenta diamond marker (Sect. 4.1). Note the relative scaling order of magnitude between variables, and the reverted x-axis for cyclonic features to aid comparison. The gradients of the lines-of-best-fit reported in the Results (Sect. 4.2) have been computed using the absolute values of the x-axis to simplify the comparison between the cyclonic and anticyclonic distributions.

The disagreement between cyclonic and anticyclonic features is further highlighted in Fig. 5, which shows the intensity distribution of anticyclonic (Fig. 5a) and cyclonic (Fig. 5b) features. Justified by our vorticity uncertainty analysis between 2017–2020, we have assumed that the negligible importance of this uncertainty can be retroactively applied to earlier years, and so Fig. 5 is representative of all features detected from 2016-2020 between 1$^{st}$ June to 31$^{st}$ October. We have chosen this temporal range to maximise the use of available data for which all four products overlap in time. It must be noted that no SSMI/S ice drift estimates in June and July of 2018 are available, due to the gap in the transition period from the SSMIS-F17 and -F18 platforms. Results show that the intensities of cyclonic features (Fig. 5b) are higher than that of anticyclonic features (Fig. 5a), as indicated by the heavier tail of the gaussian curve estimate of the cyclonic distribution. A total of 311 308 anticyclonic features were detected over this period, with the most being returned by the merged product (144 491), followed by the passive-microwave based AMSR-2 (84 171) and SSMI/S (64 263) products and the active-microwave based ASCAT (18 383) product. The number of cyclonic features returned was similar in the single-sensor AMSR-2 (105 889), SSMI/S (66 111) and ASCAT (21 991) products, while the merged product detected approximately twice as many cyclonic features (295 484) as anticyclonic features. A total of 489 475 cyclonic features were returned by all four products, 57 % more relative to the total number of anticyclonic features detected.

Inter-product comparisons indicate that there is little difference in the intensity of anticyclonic features (Fig. 5a) – as shown by the similarity of overlapping curve estimates – with only the ASCAT product (blue line) deviating slightly from the other distributions with a higher frequency of low-intensity features detected (between 0 and -0.2×10$^{-3}$ s$^{-1}$). Conversely, the cyclonic distribution curves show a larger discrepancy between products (Fig. 5b), particularly the merged product (yellow line), which shows a disproportionately small frequency of low-intensity features (between 0 and -0.2×10$^{-3}$ s$^{-1}$) and high frequency of intermediate-intensity features (between -0.25×10$^{-3}$ and -0.6×10$^{-3}$ s$^{-1}$). If the quality flag restriction is relaxed to include all the available data in the OSI-405-c product datasets, the difference between the cyclonic distributions is enhanced, with the merged product showing a much-extended tail of high intensity cyclones (not shown). This indicates that the extent of disagreement between products is dependent on the choice of quality flags used. Due to the large number of low-intensity features detected, the differences between products in detecting high-intensity features is less clear. Therefore, the most intense cyclonic and anticyclonic features for each product were compared. Here, we define the major events as those features above the 95$^{th}$ percentile (vertical dashed-lines in Fig. 5a and Fig. 5b). Similar thresholds are shown for major anticyclones in the SSMI/S (0.37×10$^{-3}$ s$^{-1}$), AMSR-2 (0.36×10$^{-3}$ s$^{-1}$) and merged (0.35×10$^{-3}$ s$^{-1}$) products, with the ASCAT (0.32×10$^{-3}$ s$^{-1}$) product again deviating slightly from the others (Fig. 5a). For cyclonic features, the merged product (-0.69×10$^{-3}$ s$^{-1}$) had the most intense threshold, followed by the AMSR-2 (-0.63×10$^{-3}$ s$^{-1}$), ASCAT (-0.62×10$^{-3}$ s$^{-1}$) and SSMI/S (-0.54×10$^{-3}$ s$^{-1}$) products (Fig. 5b). The 95$^{th}$ percentile intensity threshold was therefore 1.5–2.0 times larger for cyclonic features than for anticyclonic features.

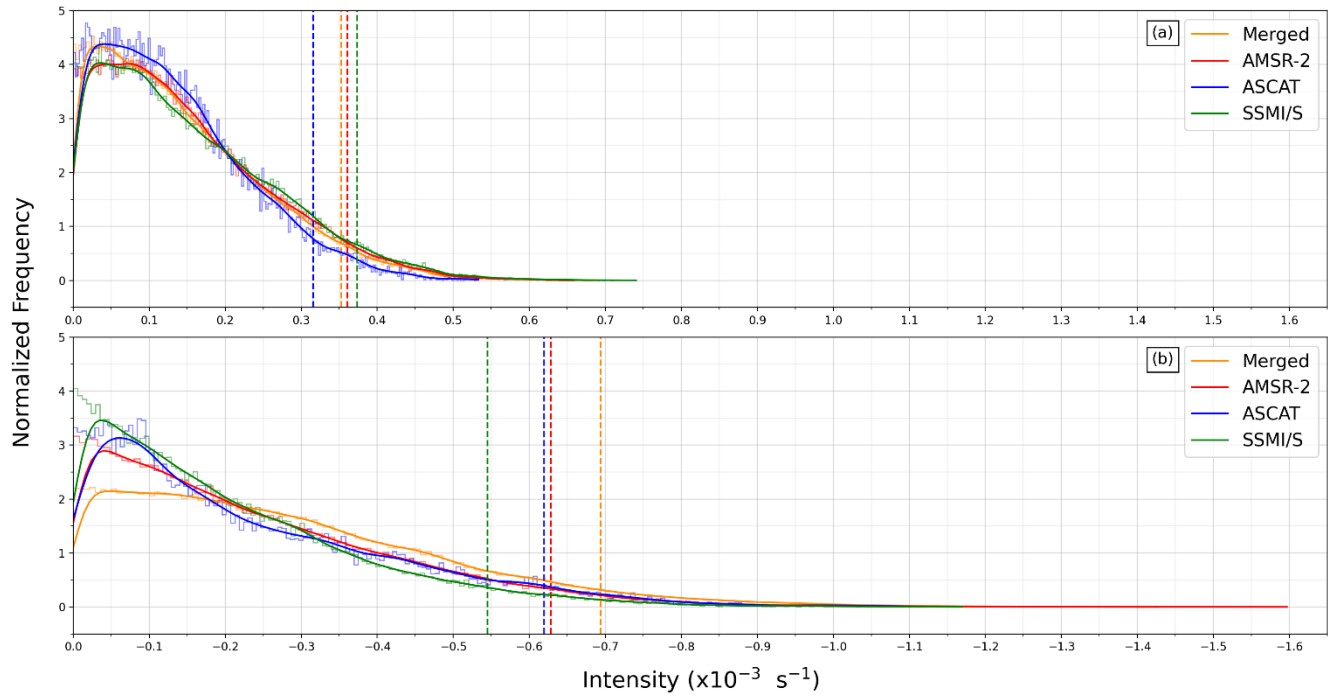

**Figure 5: The normalized frequency distribution of (a) anticyclonic vorticity features and (b) cyclonic vorticity features from 2016–2020 (note the reverted x axis for comparison). Lighter shaded lines show a stepped histogram of 200 bins, while the darker shaded curve represents a kernel-density estimate using Gaussian kernels. Vertical dashed lines indicate the intensity threshold of the 95th percentile.**

## 4.2 Year-to-year variability

The previous analysis of uncertainties indicated that the difference between the products in the number and intensity of the detected features is robust. A further analysis of the interannual variability of major events was conducted, to assess the presence of year-to-year differences during the period of data availability. The temporal range of analysis was extended to the period 2013–2020, as 2013 is the earliest year of available data in the Southern Hemisphere. However – upon defining the 95th percentile intensity thresholds like that shown in Fig. 5 – it was noticed that no cyclonic feature exceeded this threshold in 2013 or 2014 for any product. This was also the case at the 90th percentile. It is therefore noteworthy that a strict, consistent intensity threshold applied to all years does not allow us to discern the relative dynamical changes earlier than 2015. For this reason, we have defined the 95th percentile threshold for each year independently. This means that the results presented in Fig. 6 show the interannual intensity distribution of the major cyclonic and anticyclonic features based on each year's distribution. The mean, standard deviation, and intensity threshold of the 95th percentile for each year is also provided to show how different the years are and to justify the choice of this diagnostics (Table 2). Note that the AMSR-2 drift product is only available from September 2015, and no SSMI/S ice drift estimates were available in June and July of 2018. There is no obvious interannual

trend seen in anticyclonic intensity between 2013–2020 (Fig. 6a, 6c, 6e and 6g), visually indicated by the box-and-whisker rectangles remaining relatively constant in all four products and with few outlier features detected. This is further supported by the relatively small variability in the mean intensity and standard deviation between years (Table 2). Conversely, a clear cyclonic interannual trend is evident, characterized by relatively low-intensity features in 2013 and 2014, followed by an abrupt increase in intensity from 2015–2017 (Fig. 6b, 6d, 6f and 6h). From 2014–2017, the mean of the 95th percentile of cyclones increased by a factor of 2.8, 2.6 and 1.7 for the ASCAT, SSMI/S and merged products respectively (Table 2). Much like the cyclonic variability-to-intensity relationship described earlier (Fig. 4b, 4d, 4f and 4h), it is again evident that the standard deviation increases with the intensity from 2014–2017 for each available product (Table 2). All four products indicate a relatively high frequency of cyclonic outlier features – representing features which were more intense than the annual upper bound ($> |Q3| + 1.5 \times |IQR|$) – with SSMI/S product showing a larger frequency of outliers in 2015 (Fig. 6).

Table 2: The mean, standard deviation, and intensity threshold of the 95th percentile of all anticyclonic features detected between 2013–2020 (Format: Mean ± STD (Threshold)); Units in $\times 10^{-3} \, s^{-1}$). For comparison, the 95th percentile of all anticyclonic (cyclonic) features from 2016–2020 is 0.36 (-0.63) for the AMSR-2; 0.32 (-0.62) for the ASCAT; 0.37 (-0.54) for the SSMI/S; and 0.35 (-0.69) for the merged product as illustrated by the vertical lines in Fig. 3.

| ANTICYCLONIC | 2013 | 2014 | 2015 | 2016 | 2017 | 2018 | 2019 | 2020 |
|---|---|---|---|---|---|---|---|---|
| AMSR-2 | - | - | 0.43±0.07 (0.37) | 0.29±0.04 (0.24) | 0.41±0.05 (0.36) | 0.36±0.05 (0.31) | 0.48±0.06 (0.41) | 0.43±0.03 (0.38) |
| ASCAT | 0.33±0.05 (0.29) | 0.38±0.05 (0.33) | 0.40±0.09 (0.30) | 0.28±0.03 (0.25) | 0.38±0.03 (0.35) | 0.31±0.03 (0.27) | 0.39±0.04 (0.34) | 0.41±0.05 (0.33) |
| SSMI/S | 0.27±0.04 (0.22) | 0.33±0.06 (0.27) | 0.45±0.07 (0.37) | 0.31±0.05 (0.26) | 0.43±0.04 (0.39) | 0.34±0.04 (0.30) | 0.50±0.06 (0.42) | 0.42±0.04 (0.37) |
| MERGED | 0.33±0.05 (0.27) | 0.35±0.06 (0.29) | 0.43±0.06 (-0.36) | 0.35±0.05 (0.29) | 0.43±0.06 (0.36) | 0.37±0.05 (0.31) | 0.46±0.06 (0.38) | 0.41±0.03 (0.36) |
| CYCLONIC | | | | | | | | |
| AMSR-2 | - | - | -0.81±0.15 (-0.65) | -0.80±0.18 (-0.63) | -0.82±0.11 (-0.70) | -0.68±0.11 (-0.58 | -0.69±0.12 (-0.56) | -0.73±0.11 (-0.61) |
| ASCAT | -0.29±0.04 (-0.25) | -0.29±0.03 (-0.26) | -0.56±0.07 (-0.45) | -0.76±0.11 (-0.65) | -0.82±0.12 (-0.66) | -0.68±0.09 (-0.57) | -0.72±0.10 (-0.61) | -0.66±0.07 (-0.57) |
| SSMI/S | -0.39±0.05 (-0.37) | -0.45±0.08 (-0.35) | -0.42±0.11 (-0.34) | -0.58±0.10 (-0.47) | -0.77±0.12 (-0.63) | -0.60±0.10 (-0.49) | -0.66±0.13 (-0.54) | -0.68±0.09 (-0.58) |
| MERGED | -0.46±0.06 (-0.39) | -0.49±0.06 (-0.41) | -0.76±0.16 (-0.60) | -0.83±0.13 (-0.69) | -0.85±0.10 (-0.72) | -0.83±0.14 (-0.68) | -0.86±0.15 (-0.68) | -0.82±0.11 (-0.69) |

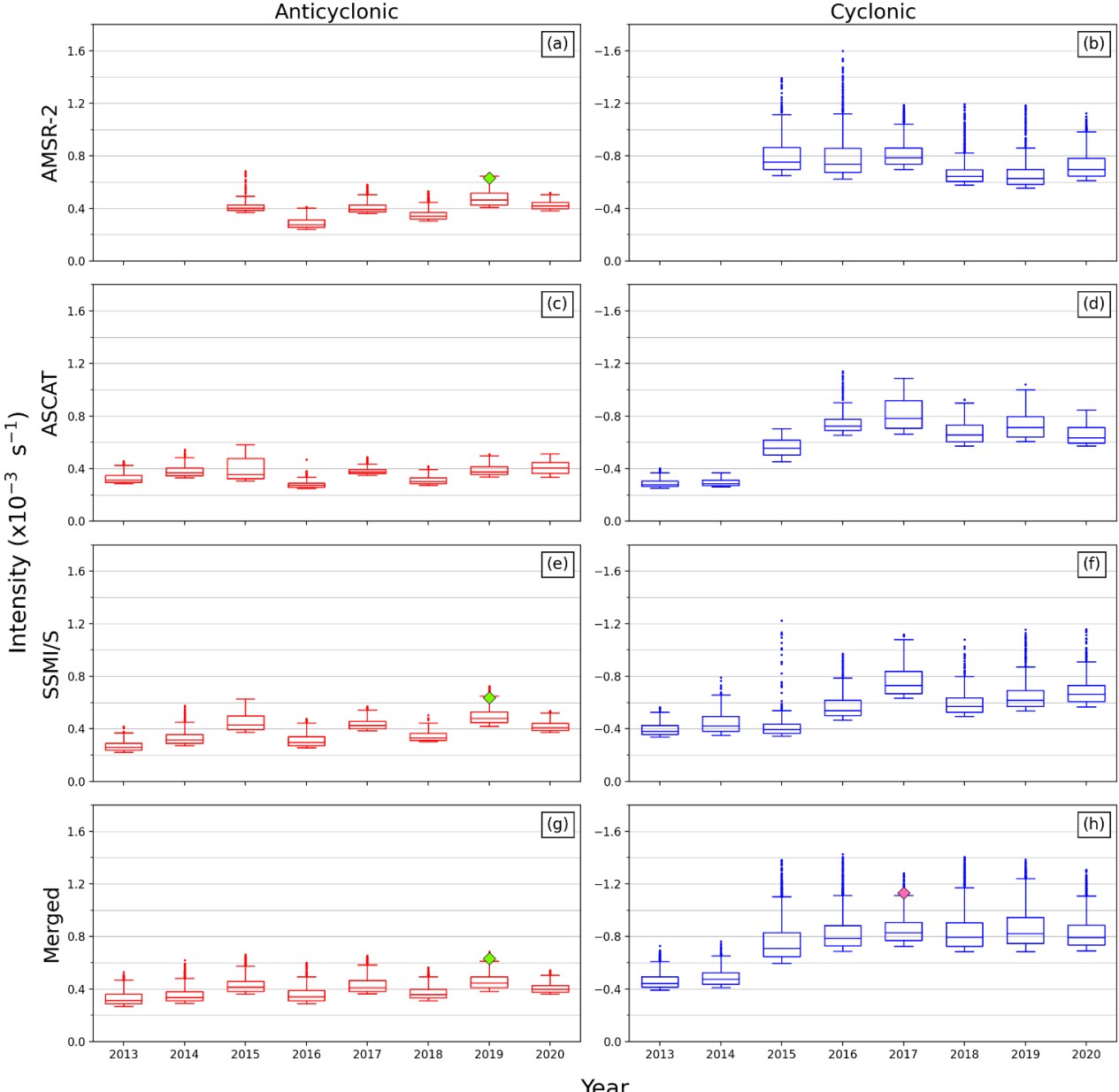

**Figure 6: The interannual distribution of the 95th percentile of features detected between 2013–2020. Cyclonic and anticyclonic features are represented with blue and red box-and-whisker plots respectively. Panels (a) and (b) show features detected using the AMSR-2 product; panels (c) and (d) using the ASCAT product; panels (e) and (f) using the SSMI/S product family; and panels (g) and (h) using the merged product. The box-and-whisker rectangles indicate the interquartile range (IQR), with the median line separating the upper (Q3) and lower (Q1) quartiles. Outlier features are shown as dots and represent features of which their intensity exceeds the upper bound (> |Q3| + 1.5 × |IQR|) or lower bound (< |Q1| - 1.5 × |IQR|). The anticyclonic feature described in Case Study 2 (Sect. 4.1.2) is shown with a green diamond marker, and the cyclonic feature described in Case Study 1 (Sect. 4.1.1) is shown with a magenta diamond marker.**

## 5 Discussion and Conclusions

This analysis presents a new method to automatically detect and quantify rotational drift in Antarctic sea ice using the EUMETSAT OSI SAF low resolution 48 h sea-ice drift product range. To our knowledge, this methodological process is the first attempt to quantify synoptic scale vorticity features in sea ice using remote sensing techniques, with the aim to establish an indicator of rotational drift in the sea ice field by which to detect current and future changes in the ice dynamics. Four products are used in this study, focusing on the Atlantic sector of the Southern Ocean: three single-sensor derived products and one merged product. Rotational features found in the sea ice may originate from both oceanic and atmospheric drivers, and while some initial studies may indicate that sub-mesoscale oceanic processes under the ice may be concurrent drivers (Biddle and Swart, 2020; Stössel et al., 2018), there is larger evidence of the role played by atmospheric cyclones in driving sea-ice motion (Vichi et al, 2019 and references therein). The case studies presented (Sect. 4.1) indicate that the vorticity field of the sea ice is strongly linked to the cyclonic and anticyclonic curvature of isobars, both in cases of extreme weather events and mild atmospheric conditions. There is no apparent evidence of oceanic drivers effecting ice rotation in this region at these spatial and temporal scales, suggesting that the detected vorticity field is dominated by weather at daily – or even sub-daily – timescales. This aligns with existing literature that sea-ice variability in the Atlantic Sector of the Southern Ocean is primarily driven by local atmospheric conditions (Kwok et al., 2017; Matear et al., 2015).

Our method is therefore oriented towards capturing sea-ice rotational events at the scales of the synoptic weather, assuming that the underlying sea-ice field would be affected at similar spatial scales. For this reason, our detection algorithm identifies circular ice drift features with a radius of 450 km ± 50 km – which is the scale of atmospheric weather and about 6–7 times the spatial resolution of the products – and quantifies the characteristics of the vorticity field within this circumference. Two case studies are presented to highlight the type of vorticity fields found in the Atlantic region and their association with atmospheric features. Case Study 1 considers an intense cyclone traversing over the sea ice, while Case Study 2 considers a persisting high-pressure cell over the ice interior. These examples underline the different quality of the drift retrieval between products, which ultimately leads to a difference in the coverage of the computed vorticity field and in the detection of features. The resulting vorticity is sensitive to the choice of quality flags used. Unsurprisingly, the merged product has the best coverage in the case of cyclones and anticyclones. This is because the merged product processes drift estimates from multiple sensors and is therefore more likely to have a good quality-flagged drift estimate at each grid point. The better coverage of good quality-flagged drift estimates from the AMSR-2 product means that its resultant vorticity field has better coverage than the SSMI/S product, despite both being passive-microwave based, while the coverage of the active-microwave based ASCAT product is considerably worse than the other three products.

This difference in coverage manifests into different vorticity fields, and so we observe a large discrepancy between products in the intensity of detected cyclonic and anticyclonic features, and major differences in the intensity distribution of cyclones.

This result is robust and significant with respect to the uncertainties in the estimation of the vorticity features as demonstrated in Sect. 4.2. The vorticity is three orders of magnitude greater than its associated uncertainty. However, it was also shown that variability in vorticity within the feature radius is of a comparable scale to its intensity (Fig. 4). The variability of the cyclonic features shows a tendency to increase with an increase in cyclonic intensity, while no such relationship is apparent for anticyclonic features. The small spread of uncertainty shown by single-sensor products in Fig. 3 contrasts with their higher

spread of spatial variability shown in Fig. 4. This, together with the high signal-to-noise ratio, suggests that the large variability detected is not a symptom of the drift uncertainty. It is also necessary to consider that the search radius of the feature detection algorithm described in Sect. 3 can be affected by contiguous cyclonic and anticyclonic features in the sea ice. Such condition would have a neutralizing effect on the value of its mean vorticity, and so these kinds of features should be represented in the low-intensity portions of Fig. 3, 4 and 5. Assuming a random distribution of these features, we expect them to be highly

heterogenous; however, only the variability of anticyclonic features detected by the merged product show higher heterogeneity in the low-intensity features (Fig. 4g). None of the products report low-intensity cyclonic features with high variability (Fig 4b, 4d, 4f, and 4h). We thus conclude that close, dipole-like features in the vorticity field are relatively uncommon, or that they are spatially more extended, and it is thus unlikely that two opposing rotation features are equally captured in the same 450 km search radius.

    The merged product displays the largest variability for both cyclonic and anticyclonic features because it combines the drift estimates from the other independent products. Our analysis of a few case studies gives some hints that the better coverage of the merged product increases the detection of rotational drift in the ice compared to the single-sensor products. The merged product has the most intense 95th percentile threshold for cyclonic features and the smallest proportion of low-intensity cyclonic

features (Fig. 5b). This is counterintuitive as it is expected that the merged product would show intermediate level results relative to its constituents, like that shown by the merged product in the anticyclonic distribution (Fig. 5a). It also shows the greatest increase in variability relative to cyclonic intensity (Fig. 4h) – illustrated by the steepest slope of its line-of-best-fit – suggesting that the higher variability detected in the merged product is linked to a larger intensity estimate. The better coverage of the merged product seemingly makes it a good candidate for synoptic scale vorticity analysis; however, we speculate that

the large variability introduced by the merging process may also cause an artificial intensification of cyclonic rotation. This is because the more extreme gradients between adjacent drift vectors in a heterogenous drift-field are manifesting into an exaggerated vorticity field. However, in the absence of independent observations that would corroborate our findings, we are unable to fully identify whether this is an artefact or a feature.

The main outcome of our analysis is that all products detect a larger proportion of high intensity cyclonic features – as indicated by the heavier tail of the cyclone distribution (Fig. 5a) relative to that of the anticyclones (Fig. 5a) – while the mean intensity of major cyclonic events is 1.5–2.0 times larger relative to major anticyclonic events between 2016–2020. Starting from the consideration that previous studies indicate that weather primarily drives sea-ice drift in the region (see Sect. 1), this suggests

that atmospheric cyclones may inject more rotational momentum into the underlying sea ice than anticyclones. Furthermore, the Weddell Sea is dominated by a climatological low-pressure cell – termed the Weddell Low – which is the result of the frequent passing of cyclones through this region caused by intense cyclogenesis in the Atlantic Sector (Grieger et al., 2018; Simmonds et al., 2003; Wei and Qin, 2016) or low pressure features crossing the Drake Passage to the east (Gonzalez et al., 2018). Since it has been shown that the ice vorticity field is primarily weather driven, this dominance of cyclonic rotation in the atmosphere likely contributes to the more frequent and intense cyclonic vorticity features detected in the sea ice. However, there may be other factors that could lead to a higher proportion of cyclones in sea-ice drift. The feature-tracking method of drift retrieval may be susceptible to inaccuracies under conditions of rapid dynamic and thermodynamic changes in sea-ice properties, such as in the event of a strong cyclone traversing the sea ice where the motion field has strong temporal gradients. This is evidenced in Case Study 1, where an atmospheric cyclone's tendency to modify the radiometric properties of the sea ice seemingly hinders the drift retrieval process, thereby resulting in poor quality-flagged drift estimates from the affected locations. Conversely, high-pressure cells are typically characterized by calmer atmospheric conditions, where the local weather is less influential on the quality of the drift estimates. This may explain the larger disagreement between products in the distribution of cyclonic features (Fig. 5b) than that of anticyclonic features (Fig. 5a). It is therefore necessary to consider that rapidly moving ice floes under the effect of polar storms may be blurring the rotational drift we are attempting to estimate over a 48 h period, causing a larger discrepancy between products and contributing to the large spatial variability observed in cyclonic features. It is thus difficult to discern whether the dominance of cyclonic rotation in the ice – both in frequency of events and their intensity – is due to 1) the dominance of cyclonic rotation in the overlying atmosphere; 2) atmospheric cyclones being more effective at engendering rotational motion in the underlying sea ice than anticyclones; 3) the feature tracking method of drift retrieval being overly sensitive to the conditions under an atmospheric cyclone; 4) any combination of the aforementioned considerations.

Despite the differences between products, our results give a consistent indication that anticyclones are less intense than cyclones, and that there is a change in the intensity of the latter after 2015. This gives us confidence in the method and allowed us to perform a provisional analysis of the interannual variations in the most intense cyclonic and anticyclonic features from 2013–2020. We are cognisant that this period is too short to detect any climatic signal, and therefore this analysis is meant to demonstrate the use of this methodology to detect possible trends in the future. The results show that major anticyclonic events have remained relatively constant from 2013–2020 according to the merged, ASCAT and SSMI/S products. The AMSR-2 detected the same uniformity since its derived drift product became available in September 2015. Conversely, a substantial change in the interannual distribution of major cyclonic events is evident, where all available products detected an abrupt increase in their intensity from 2014–2017. This increase coincides with the record decline in the Antarctic sea-ice extent observed from late winter 2015 (Parkinson, 2019; Turner et al., 2017). Furthermore, there is also an increase in the number of outlier features per year from 2015 onwards, suggesting that the most intense cyclonic features may have been intensifying further in the last years on record. Among other causes that involve atmospheric and oceanic components (Blanchard-

Wrigglesworth et al., 2021; Meehl et al., 2019; Stuecker et al., 2017; Wang et al., 2019a), it has been argued that an intensification in polar storms in 2016 contributed to the anomalously quickened SIE decline from 2015, as the overlying
winds of these synoptic features induced changes in the sea-ice dynamics (Wang et al., 2019b).

Our results show that sea ice in the Atlantic sector was more susceptible to cyclonic rotational features after 2015, which can be interpreted as a response to an increased incidence of polar cyclones. If the sea ice was thinner and more prone to free-drift motion in general, then we would expect both an increase in cyclonic and anticyclonic rotation, but instead, only an increase
in the intensity of cyclonic features is detected. This indicates that the increased incidences of polar storms are likely injecting more momentum into the underlying sea ice, however the extent of which is difficult to quantify given the disagreement between satellite products. Prior to performing further analysis of drift variability and longer-term trends linked to polar atmosphere variability, further validation of the vorticity metric with *in situ* experiments is required to better discern the differences between products. We therefore argue for the need of a concerted experiment to increase the number of
observations of sea-ice drift in Antarctic sea ice to assess the quality of the products and their use to quantify rotational features. A better understanding of these features will enable us to confidently use rotational drift of sea ice as a potential derived index to detect climatic trends.

**Data and code availability**

The OSI-405-c product FTP server is available at ftp://osisaf.met.no/archive/ice/drift_lr/. The ECMWF ERA-5 atmospheric
reanalysis data is available at: https://cds.climate.copernicus.eu/cdsapp#!/dataset/reanalysis-era5-single-levels?tab=overview. The python script to compute and extract the cyclonic and anticyclonic sea-ice features from the OSI-405-c 48 h datasets, as described in Sect. 3, is available at: https://github.com/waynedejagerUCT/Rotational_Drift_From_Space.git.

**Author contributions**

Authors W. de Jager and M. Vichi contributed equally to this manuscript.

**Competing interests**

The authors declare that they have no conflict of interest.

**Acknowledgements**

This work has received funding from the National Research Foundation of South Africa (South African National Antarctic Programme, Project No. 118745).

We would like to acknowledge and give thanks to our reviewers: Dr Thomas Lavergne, Dr Valentin Ludwig and an anonymous third reviewer, for their time invested into our work. Your comments and suggestions have helped us to improve the quality of our paper and to expand beyond the initial brief communication. Your efforts are greatly appreciated. Furthermore, thank you to our handling editor, Dr Christian Haas.

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
