# Peer review of "Rotational drift in Antarctic sea ice: pronounced cyclonic features and differences between data products"

_The Cryosphere, 2021_

## Referee Comment (RC1)

**Review of tc-2021-144: Lack of agreement in remote sensing detection of cyclonic drift caused by Atlantic weather in Antarctic sea ice.**
**By: W. Jager and M. Vichi**

The authors are interested in discriminating the relative contribution of thermodynamic and dynamic processes to the year-to-year variability and trends of the sea-ice cover in the Southern Ocean. Since satellite-based sea-ice concentration data alone do not allow such partition, they look at satellite-based sea-ice drift fields and specifically the detection and quantification of cyclonic and anticyclonic drift patterns in the Atlantic sector of the Southern Ocean. They access four sea-ice drift products from the EUMETSAT OSI SAF (one from ASCAT, one from AMSR2, one from SSMIS, and a multi-sensor product combining the first three) and report large discrepancies between the relative vorticity fields computed from these four products, both in the annual distributions and the general distribution of the intensity features detected. The authors conclude that the observed disagreement between the satellite-based products impede further analysis of the partition of dynamic and thermodynamic contributions to the evolution of the sea-ice cover.

This manuscript is framed as a short communication in which the authors report on a negative result: the discrepancy between the relative vorticity fields derived from satellite-based sea-ice drift products is such that these cannot be used for the intended purpose. The satellite-based products should be made more consistent, the vorticity results be more similar, before they can be used in further scientific analysis. The satellite-based products are not reliable enough for this type of analysis.

I have a major comment about this manuscript (leading to requesting additional analysis or at least adding a detailed discussion), and a series of more minor comments.

**Major comment:**

No satellite-based product is a perfect measurement. The noise in the raw satellite observations and the uncertainties introduced by the retrieval algorithms all contribute to a retrieval noise. In addition, differences in timing of the various satellites orbiting the Earth can result in representativity uncertainties: the same algorithm applied to different satellite missions will result in different geophysical fields just because the timing of the observations are different. The EUMETSAT OSI SAF sea-ice products do provide numerical estimates of these uncertainties in the product files (since June 1$^{st}$ 2017) and has conducted validation against buoy data (see the validation reports). The uncertainties in the components of the sea-ice drift vectors will naturally propagate into uncertainties of the vorticity metric. How the drift uncertainty propagates into the vorticity metric can be estimated theoretically (Dierking et al., 2020) or numerically e.g. using Monte-Carlo simulations. An analysis of the propagation of uncertainty from the uncertainty in the drift components to the vorticity metric, and how the uncertainty on vorticity relates to the differences observed between the four products is missing. For example, do the 4 vorticities agree within their error bars, or are they really returning "no agreeable pattern" as is stated (page 5, line 132)? In other words: are all 4 products seeing the same signal but with a lot of noise (the products are collectively inadequate for this application), or are they seeing different signals (in which case some of the products can be better OR the timing and spatial coverage differences have a large impact on the vorticity)? Without this analysis, we cannot put an error bar on the retrieved vorticity values, and cannot discuss if one of the satellite-based product is more reliable than the others for estimating vorticity. This would however have been an interesting results of this study: if one of the product leads to a lower uncertainty on the vorticity, then this product could potentially be used for later analysis (it would be a more useful conclusion than stating that none of the products are usable).

This is also relevant when it is observed that "the processing chain used in the development of the multi-sensor merged ice drift product can induce additional rotational energy into the resultant vector field". It is known (e.g. from the validation reports produced with the OSI SAF sea-ice drift products) that the multi-sensor product is "smoother" (in space) than the single-sensor products, because it reduces the retrieval uncertainties but also act as an averager of the daily sub-drift variability (captured by the single-sensor products at different observation times). It is thus maybe not surprising that the multi-sensor product shows higher "rotational energy" and it might very well be that it is the more correct product (your phrasing suggests that energy is added by the algorithm, while it would rather be that the single-sensor products miss some of the energy because they are more noisy).

In summary, I recommend that an analysis of the propagation of uncertainties to the vorticity metric is performed and added to the manuscript. The OSI SAF sea-ice drift products have documented the uncertainties they have on their drift components (through quantitative uncertainties and validation against buoys), and these should be used to tell 1) are the differences in vorticity observed between the products just the expression of a noise or are they seeing actually different vorticity patterns, and 2) is one of the products more accurate than the others in terms of vorticity, also concerning the "additional rotational energy" claim. If an uncertainty propagation analysis cannot be conducted, it is strongly recommended that a thorough discussion is added about the significance of the documented uncertainties on sea-ice drift components on the conclusions (again also concerning the "additional rotational energy").

**Other comments and questions:**

Page 2, line 34: Isn't SIC rather defined as the proportion of ice-covered water to total area (ice-covered or not).

Page 4, line 111: "horizontal and vertical components" this refers to a 2D project map with components along the vertical and horizontal directions of the grid, but can be mis-read as the vertical (3D) component of the sea-ice drift. You mean "x and y axis of the grid".

Page 4, Methodology question 1: how are missing vectors dealt with for the single-sensor products: ASCAT has many missing vectors especially at lower latitudes (outskirt of the domain), the multi-sensor product has many more. How is the vorticity computed in case a vector is missing?

Page 4, Methodology question 2: did you use all the vectors, irrespective of their status_flag, or did you remove some of the more dubious flags?

Page 4, Methodology question 3: Do the subdomains Dr overlap? Specify in the text. If yes, the vorticity events thus contributed several times?

Page 4, Methodology question 4: At the beginning of section 4 you refer to the "intensity distribution … identified by the algorithm..." but your section 3 Methodology does not clearly define an "intensity". Add this in section 3, or be more specific in section 4.

Page 4, Methodology question 5: "Any subdomain with a mean vorticity of zero is ignored". It seems unlikely that the mean vorticity would return exactly 0. Is you test against 0 exactly, or within a range around 0 (what range?). If exactly 0 it could be worth stating in which (frequent) conditions the vorticity is exactly 0.

Page 4, Methodology question 6: Have you looked at the intensity of "significant" cyclonic and anti-cyclonic events (intensity of events above a vorticity threshold)? It could indicate if the difference you observe build from low-signal / noise events, and if the products agree better on the

major events (that are possibly more relevant to the original objective of partitioning the dynamic vs thermodynamic contributions)?

Page 8, line 181: Before stating that there is a "a large discrepancy between products", we need to quantify what "large" is wrt the uncertainties: is it just noise or really discrepancies (observing different signals).

Page 8, line 198: Here again, the merged product is described as detecting a "disproportionaly large frequency" but what if it is the most accurate of the four, and it is the larger noise in the 3 other products that leads to an underestimation of the high intensity features?

Page 8, first lines: Here again, what is the impact of the multi-sensor product having fewer missing vectors than the 3 single-sensor products?

**Typos and editorial suggestions:**

Page 3: line 71: missing "is" (it is necessary).

Page 3: line 74: consider having "detection" before "quantification" (1st detect, then quantify?)

Page 3, line 78: maybe "shared" or "common" would be better than "unique"?

Page 3, line 85: "range" → "family"

Page 3: line 93: The multi-sensor merged product does more than "treats missing data...", suggest replacing "treats missing data by means of a two-step" with "implements a two-step"

Page 4, line 113: "dx and dy are the grid spacing (62.5km)".

Page 5 Figure 1: suggest to write the product name (ASCAT, AMSR, etc…) in the plot area in addition to of (a), (b), etc…

Page 6 Figure 2: same suggestion as for Figure 1.

**References:**

Dierking, W., Stern, H. L., and Hutchings, J. K.: Estimating statistical errors in retrievals of ice velocity and deformation parameters from satellite images and buoy arrays, The Cryosphere, 14, 2999–3016, https://doi.org/10.5194/tc-14-2999-2020, 2020.

---

## Referee Comment (RC2)

**Review of paper "Brief communication: lack of agreement in remote sensing detection of cyclonic drift caused by Atlantic weather in Antarctic sea ice"**

**1 Content**

The authors de Jager and Vichi present a brief study on the intercomparison of four sea-ice drift datasets, namely the OSI-SAF merged drift product OSI-405-c and its three constituents. These are drift products derived from AMSR2, ASCAT and SSMI/S. The study focuses on the detection of cyclonic and anticyclonic rotation in the Atlantic sector of the Antarctic pack ice zone. For the years 2015-2020, the most intense cyclonic and anticyclonic features of each 48-hour period are compared statistically. The authors find that there is stronger cyclonic than anticyclonic vorticity. Comparing the products, the authors report that the merged product shows more high cyclonic vorticity values than each of the input products. This is interpreted by the authors as additional energy introduced by the merging scheme.

**2 General comments**

The paper is well-written and mostly easy to comprehend. The results are enough, both concerning relevance and quantity, to warrant publication as brief communication. My main point of criticism is that the paper is rather short on the discussion of the results. The finding that the merged product shows more high-intensity features than the single-sensor products is really interesting, but the authors do not give a reason. I understand that an in-depth analysis is beyond the scope of a brief communication, but I would like to see which ideas the authors have for further research, so that a follow-up study could build upon their work. The following questions came to my mind:

▷ What is the reason for the above-mentioned mismatch between the merged product and the single-sensor products?

▷ How do you judge this mismatch? Is it an artifact arising from the merging method or does it bring additional insight which the single-sensor products can not provide?

I would ask the authors to elaborate on this in their Discussion section. Also, it would be good to know if similar results have been achieved by other studies.

**3 Specific comments**

**3.1 Abstract**

L11: Concerning the word "alternative": A bit misleading because it sounds as if the concept of sea-ice extent (SIE) would be used to quantify changes in sea-ice dynamics. I suggest to leave out "alternative".

L18: For me, the processing chain is merely the technical implementation of the merging method, therefore I would suggest to refer to the merging method here instead of the processing chain.

L18/19: I suggest to add that only cyclonic momentum is added.

**3.2 Introduction**

General: Should mention that, unlike in the Arctic, SIE in the Antarctic was quite constant until recently

General: When describing your motivation, you might also want to mention more explicitly that we expect increased sea-ice drift in future, given the thinning of sea ice and the increased storminess.

L29: Do "scarce" and "sparse" not effectively mean the same thing?

L36: I would suggest to replace "ice edge" by "marginal ice zone", there is seldom a sharp and abrupt transition between sea ice and ocean which would justify the term "edge"

L38: Why is the variability dramatic? I would suggest something more objective and less drastic like "high" or "pronounced"

L39-45: Talking about limitations of SIE, you might also want to refer to Notz (2014)

L56-58: I suggest to restructure and split the sentence: "Ice movement is primarily driven by . . . . Other factors are waves, ocean tilt. . . ".

L68: Please provide a reference for your statement that the Southern ocean hosts some of the most energetic storms worldwide.

L69: MIZ has not been defined yet.

L71: Much has changed since 2003/2004, please provide more up-to-date references. Also, "it therefore" should be "it is therefore".

L74: Instead of "daily timescales", you could be more specific and speak of "two-daily resolution"

L74: What exactly is the method which you propose? Taking the maxima and minima of the vorticity within the domain as described in L114-L120? Woud be good to state this more clearly, to me it was not immediately clear although it was the initial motivation for your paper.

L78: As outlined above, I doubt that the term "processing chain" is appropriate here and suggest to replace it by "merging method" or something similar

L82-83: Isn't your conclusion that the merging introduces additional cyclonic rotation?

In this case, you can also write this here instead of using the weaker formulation ". . . can induce additional. . . ". Also, you could specify already here that the additional rotational energy comes from cyclonic rotation.

**3.3 Data**

General: Please provide references for the single drift products. Also, it would be good if you can state here which region and months you use.

L90: What is meant by "SSMI/S instrument range"? Please specify.

L94: weighted by what?

L97: I think "coarse" would be more appropriate than "large" when speaking of resolution.

L97: Can you comment on the typical size of the cyclones which you detect in relation to the grid spacing of 62.5 km? Be careful to not mix up grid spacing and resolution.

L101: Please specify the projection (NSIDC projection with the latitude of true scale at 70°S?) or give a reference.

**3.4 Methodology**

L105: The readability here and in the rest of the paper would be better if you could adopt the practice to use "sea-ice" when speaking about sea-ice properties (sea-ice vorticity, sea-ice concentration etc) and use "sea ice" when referring to it as a noun.

L107-110: Domain and months should be specified in the Data Section. It would also be good if you could state how large the area is in $km^2$. What is your criterion for the "ice-covered area"? SIC above 15 %? Please state this here.

L115: If you choose the maxima/minima of the mean vorticities, you might get into trouble if there are outliers which are not representative of typical cyclonic/anticyclonic features. Can you comment on this? Did you compare the results which you get by taking the extreme values to the results which you would get if using a more robust estimator like the 95th percentile? Would you expect differences arising from this? Please briefly discuss.

**3.5 Results**

General: Please state how many data points there were per year. Was it always the same number?

Figure 1: If your main goal is to compare the products among each other, it might make more sense to have one panel per year instead of one panel per product. If it does not overload the plots, you could also consider merging the panels a–d to one and mark the products by different colors. With the current alignment, I find it hard to compare the results of single years between sensors. Same for Fig. 2.

L124-126: This technical description of the box-and-whisker plot could be moved to the

caption of the Figure.

L131: This is the kind of statement which is hard for me to assess if the four ice motion products are shown in separate panels. Can you give the values to which you refer here in a Table?

L132ff: What exactly do you refer to by "spread"? Interquartile range? Range between whiskers? Further, it would be good to also at least mention the actual magnitude of the cyclonic features, not only the spread, even if the latter is your main focus.

L139: What do you mean by "high levels of interannual variability"? I do not have the impression that the medians or interquartile ranges in Fig. 1b vary more than in a, c or d.

L141: "which being detected": Should this be "which was detected"?

L144: Should mention the reduced y-axis range of Fig. 2 compared to Fig. 1.

L170-176: Please put the IQR and $\sigma$ values in a Table, this would be much easier to grasp and would improve the readability.

**3.6  Discussion and Conclusions**

L187: "...increasing trend...": Do you refer to the spread or to the absolute values of vorticity?

L187: Please discuss the robustness of this trend, given that your study period is quite short. Is this also found by other studies?

L197-202: Very interesting indeed to see that the merged product shows more vorticity than any of the others. I would like to see this discussed in more detail. An in-depth discussion would probably be too much, but can you elaborate on potential reasons or give directions for future research? Also, do you trust this result? Would be good to get an idea whether the additionally introduced rotation is valuable information which we can not get from the single-sensor observations or whether it is an artifact of the merging.

L204-205: Please give a reference or explain why you expect disproportionately high frequency of low-intensity features in the Eastern Weddell Sea.

L209: See my comment to your L74. Please describe the method briefly, since it was the main motivation for your work.

L212-219: Please give directions/ideas how to find out the reason for the mismatch in the cyclonic drift features.

**References**

D. Notz. Sea-ice extent and its trend provide limited metrics of model performance. *The Cryosphere*, 8(1):229–243, 2014. doi: 10.5194/tc-8-229-2014. URL https://tc.copernicus.org/articles/8/229/2014/.

---

## Referee Report (RR1)

**Re-review of paper "Rotational drift in Antarctic sea ice: pronounced cyclonic features and differences between data products"**

**1 Content**

The authors de Jager and Vichi re-submitted a paper which analyses the differences between different satellite-based sea-ice drift products from EUMETSAT OSI-SAF (product OSI-SAF 405 c), namely five single-sensor-based products (AMSR2, ASCAT, 3x SSMI$x$) and a merged product based on the formerly mentioned ones. Much content has been added to expand the manuscript from the previously submitted "Brief communication" to a full-fledged research article. The major differences are the inclusion of an uncertainty estimate and analysis, the discussion of spatial variability and the restriction of the analysis to high-quality data as per the quality flags of the drift product.

**2 General comments**

I appreciate the effort which the authors made to submit this expanded version. I fully agree with their decision to extend the manuscript to a full research article, following the handling editor's recommendation. I enjoyed reading this manuscript and found my concerns from the first round of review addressed satisfactorily. However, I still have some comments which I would like to see addressed before publication. Line numbers in my specific comments correspond to the track-change version of the revised manuscript.

**3 Specific comments**

**3.1 Abstract**

L14: " . . . Antarctic ice characteristics.": You might consider removing "Antarctic", since the need for methods to quantify sea-ice changes is also there in the Arctic.

**3.2 Introduction**

General: I find the Introduction quite long. The part after L72 was interesting and well-written, but the part before could for my taste be condensed. I suggest to build your

motivation on the increased storminess and the recent changes in Antarctic sea ice, but not so much on having an additional measure for assessing climatological trends as a complement to SIE.

L44-45: A reference would be good, for example IPCC AR6.

L72: You start your sentence with "Studies...", but then refer to only one. I suggest to either rephrase or add more references.

L77: "El Nino" $\rightarrow$ "El Niño"

L85: Replace "ice floes" by "ice pack"

L96: I suggest to use present tense throughout the paper whenever possible, especially when describing your own work.

L102: Please mention that all products are from OSI-SAF.

L105: "assess" $\rightarrow$ "to assess"

L106: Please specify the period by saying which years you refer to.

**3.3   Data**

Generally: Please spell out the acronyms when you use them for the first time.

L123: Is it intentional that you write "SSMIS-f17" or should it be SSMIS-F17 (also in the following lines)?

L125:"SSMI/S" $\rightarrow$ "SSMIS"

L136: Please provide a reference (URL) for the NSIDC projection.

L139: A reference for the limitations would be nice.

**3.4   Methodology**

L145: "SSMI/S" $\rightarrow$ "SSMIS". Please also check for other occurrences in the paper.

L152: "Acceptable" sounds quite subjective. Readers who are not familiar with the meaning of the single flags (such as myself) might wonder why you choose flag 20 as criterion. Please give very briefly some information about the meaning of the flag numbers, and your reason to choose flag 20 as threshold.

L159-161: I wondered if there may be cases where there is one cyclonic and one anticyclonic feature in $D_r$. If yes, taking their mean would erroneously return comparably small rotational activity. Can this happen? Please elaborate on this in the Discussion.

L168: 1) "resolution" should be "grid spacing". Also, I suggest to simply write "...a grid spacing of 62,5 km is fine enough to capture these features.", the current formulation sounds a bit odd.

L172: Dierking et al. (2020) is missing in your reference list.

Equation 2: $\Delta T$ is probably constant, right? Does this mean that $\sigma_{vort}$ scales linearly with $\sigma_{tr}$? Please mention this briefly.

General: How is the merged product's uncertainty calculated? As a Gaussian error propagation based on the other three product's uncertainties? Please describe this.

**3.5 Results**

Figs 1&2: These are important Figures, but it would be really helpful to use a density plot or 2d histogram instead of a simple scatter plot. With this amount of points, it is visually impossible to see the actual distribution of values. This would also help to assess the reliability and representativeness of the trend lines. Can you provide some estimation of the robustness of the fit lines, for example a standard error of the linear regression? Also, it would be good to add the number of points, the slope and the intercept, either in the Figures themselves or in the caption.

General: I was a bit surprised about the discrepancy between AMSR2 and SSMIS. Since they are both passive-microwave based, I would have expected them to be closer to each other than to the active-microwave based ASCAT data. Can you elaborate on this at some point?

L218: What do you mean by "gradient factor"? The slope of the fit line? If yes, I would suggest to call it "slope".

L237: "intensity" → "intensities"

L238-245: Are these the same features that went into Figures 1 and 2? If yes, you could consider to add the numbers of features either to Table 1 or to the Figure captions. Also, why does AMSR2 detect so much more features than SSMIS?

L261: "...than _for_ anticyclonic features."

L270: "into" → "of"

L278: What is meant by "statistical " mean?

L279: AMSR2 is in use since late 2012, it is the drift product which is available since 2015.

**3.6 Discussion and Conclusions**

L308: Please describe briefly what is new about your method.

L311: Would be more consistent to write that you use four products.

L317f: From this formulation, it sounds like the 450 km would be 6-7 times the scale of atmospheric weather. I suggest to reshuffle the sentence between the dashes and write " which is the scale of atmospheric weather and about 6 - 7 times..." to be clearer.

L321ff: Can you speculate on possible physical or methodological reasons for the discrepancy between a) the single-sensor products and b) the single-sensor products and the merged product? Having an idea about this would add much relevance to the study and help it to be even more than a description of an interesting finding.

L341f: It is not entirely clear to me why more variability would artificially intensify cyclonic rotation. Please explain in more detail.

L343: "observation" → "observations"

L352: See comment to L279.

L357-359: Are all the references needed? if not, please pick the two or three most relevant ones.

General: It would add much value if you could give the reader a recommendation in which cases the merged product is more appropriate and in which cases one might be better off with one of the single-sensor products.

**3.7   Code**

Thank you for including the code for the analysis. It would be good, however, if you could add a small readme file to the repository which instructs the user how to use the code. Even though the code is plain and well-written, I think that it would be of value if you could add some comments, so that potential users can use it right away.

---

## Author Response (AR4)

**TC-144-2021: Response to reviewers of Manuscript Version 1**

**Overview of our revised manuscript after major changes:**

Following the suggestions made by the reviewers – as well as the handling editor's suggestion to expand the paper beyond a brief communication – we have modified and added a considerable amount to our revised version. Due to the significance of the new results and our intention to provide a logical flow in communicating our results, we have also changed the ordering of some of our previous results as described in the following.

Firstly, we have added a further component to our revised manuscript which investigates the role of uncertainty propagation in the computation of the vorticity, as this was a major suggestion made by Reviewer 1 (Sections 3 and 4.1). To do this, we ignored drift estimates corresponding with the rejection quality flags (i.e., flag values 0-20 in the OSI-405-c products), and only used those with a quality index flag 20-30. This decision was taken to ensure that there is sufficient ice-covered area to perform our analysis for all the products. While we acknowledge that quality flags 20, 21 and 22 may be of degraded quality, the displacement uncertainty of these flags is made available in the dataset (as of 1st June 2017) and thus we can quantify this potential noise. The method of our uncertainty propagation can be found as described in Section 2.4 of Dierking et al. (2020). To reduce the influence of using non-nominal quality flags, we decided to redefine our period of analysis to between 1st June – 31st October (previously was 1st April – 30th November). This was decided because the freezing months have a higher proportion of nominal-quality drift estimates. Furthermore, the initial manuscript showed results between 65° W and 10° E, but the revised manuscript shows results between 65° W and 50° E. This change was made to reduce the influence of the region boundary 10° E, an area where sea-ice coverage band is wide. The sea-ice coverage at 50° E is thin relative to the scale of features, and so no features are detected here because of the insufficient ice cover, and not by an artificial boundary parameter. We found that the uncertainty of the vorticity metric was 3 orders of magnitude smaller than the mean vorticity of the same area (revised Fig. 1), and so with these results we can assess trends in feature intensity more confidently.

Furthermore, the spatial variability of vorticity measurements within the circumference of the detected features was also analyzed. This was done to contrast the relative significance of the feature's uncertainty compared to its variability, and it was found that the variability is of the same order of magnitude as the feature intensity (revised Fig. 2), and so should be considered when investigating trends.

We then expanded our analysis to include year 2016-2020. This decision was made to maximize the period of overlap between products, mostly limited by the September 2015 launch of the AMSR-2. All features (not just the daily maxima and minima) in this period are then compared for both cyclonic and anticyclonic features (revised Fig. 3). Here we see both differences between cyclonic and anticyclonic distributions, and the same (though reduced) discrepancy of cyclonic distributions between products that was reported in our first submission.

 Finally, it was suggested by both reviewers that we consider an analysis of the most 'significant' features. We agreed that this would be a useful addition to our revised manuscript. To do this, we

defined our "major features" as the 95th percentile of most intense features independently for both cyclonic and anticyclonic features, where the intensity of a feature is defined as the mean vorticity of the area within its circumference. Figure 3 now includes a visualization of the percentile threshold. We then extend our analysis back further to include all available OSI-405-c drift estimates, which starts in 2013 for the Southern Ocean. However, when analyzing the interannual variability, it was noticed that no major cyclonic features from 2013 or 2014 exceeded the 95[th] percentile threshold (similarly for the 90[th] percentile too). This meant that using a fixed percentile threshold was not an effective means to communicate the results of the trend analysis. We therefore decided to quantify a 95[th] percentile threshold independently per year. This means that the results discussed here do not describe the interannual variability relative to a fixed-in-time rotational drift intensity, but rather how the most intense features of each year vary relative to that of other years. This decision allowed us to show the pronounced change in the distribution of the most intense events of each year. More specifically, it is shown that there is little change in the distribution of major anticyclonic events, but substantial change in the distribution of major cyclonic events (revised Fig. 4). There is a rapid increase in the intensity of the major events from 2014-2017.

The results of our revised version concluded that:

1. The uncertainty of the vorticity metric is very small when using the OSI-405-c product.
2. There is a dominance of cyclonic drift detected by all products, but the products indicate different distributions. This discrepancy is particularly large when using the merged product, which is characterized by the most intense cyclones. Given our analysis of the uncertainties and variability in relation to the intensity of the events, we speculate that this may be a consequence of the increased number of features introduced during the merging and 48-hour detection window.
3. There is an abrupt increase in cyclonic drift between 2014-2017, which we potentially attribute to the increased polar storms reported in the literature during this period.

**Response to Reviewer 1**

Dear Dr Thomas Lavergne

Firstly, I would like to thank you for the time you have given to provide us with constructive suggestions to improve the quality of our paper. Your efforts are very much appreciated.

In this document we have provided an overview of the major changes we have made for our revised manuscript after acting upon the suggestions made by you and Reviewer 2. We have also responded to each of your specific comments, and those of Reviewer 2, in a point-by-point format. Your original comment appears in *italics*, while our response appears in **bold**.

**Responses to your specific comments:**

*Major comment: No satellite-based product is a perfect measurement. The noise in the raw satellite observations and the uncertainties introduced by the retrieval algorithms all contribute to a retrieval noise. In addition, differences in timing of the various satellites orbiting the Earth can result in representativity uncertainties: the same algorithm applied to different satellite missions will result in different geophysical fields just because the timing of the observations are different. The EUMETSAT OSI SAF sea-ice products do provide numerical estimates of these uncertainties in the product files (since June 1st 2017) and has conducted validation against buoy data (see the validation reports). The uncertainties in the components of the sea-ice drift vectors will naturally propagate into uncertainties of the vorticity metric. How the drift uncertainty propagates into the vorticity metric can be estimated theoretically (Dierking et al., 2020) or numerically e.g. using Monte-Carlo simulations. An analysis of the propagation of uncertainty from the uncertainty in the drift components to the vorticity metric, and how the uncertainty on vorticity relates to the differences observed between the four products is missing. For example, do the 4 vorticities agree within their error bars, or are they really returning "no agreeable pattern" as is stated (page 5, line 132)? In other words: are all 4 products seeing the same signal but with a lot of noise (the products are collectively inadequate for this application), or are they seeing different signals (in which case some of the products can be better OR the timing and spatial coverage differences have a large impact on the vorticity)? Without this analysis, we cannot put an error bar on the retrieved vorticity values, and cannot discuss if one of the satellite-based product is more reliable than the others for estimating vorticity. This would however have been an interesting results of this study: if one of the product leads to a lower uncertainty on the vorticity, then this product could potentially be used for later analysis (it would be a more useful conclusion than stating that none of the products are usable).*

*This is also relevant when it is observed that "the processing chain used in the development of the multi-sensor merged ice drift product can induce additional rotational energy into the resultant vector field". It is known (e.g. from the validation reports produced with the OSI SAF sea-ice drift products) that the multi-sensor product is "smoother" (in space) than the single-sensor products, because it reduces the retrieval uncertainties but also act as an averager of the daily sub-drift variability (captured by the single-sensor products at different observation times). It is thus maybe not surprising that the multi-sensor product shows higher "rotational energy" and it might very well be that it is the more correct product (your phrasing suggests that energy is added by the algorithm, while it would rather be that the single-sensor products miss some of the energy because they are more noisy).*

*In summary, I recommend that an analysis of the propagation of uncertainties to the vorticity metric is performed and added to the manuscript. The OSI SAF sea-ice drift products have documented the uncertainties they have on their drift components (through quantitative uncertainties and validation against buoys), and these should be used to tell 1) are the differences in vorticity observed between the products just the expression of a noise or are they seeing actually different vorticity patterns, and 2) is one of the products more accurate than the others in terms of vorticity, also concerning the "additional rotational energy" claim. If an uncertainty propagation analysis cannot be conducted, it is strongly recommended that a thorough discussion is added about the significance of the documented uncertainties on sea-ice drift components on the conclusions (again also concerning the "additional rotational energy").*

**As mentioned in the above overview, we have included a uncertainty propagation in our analysis. The uncertainty was estimated theoretically in Sec. 2 following Dierking et al. (2020) as suggested by the reviewer and assessed the results in Sec. 4.1 in the new Fig. 1. We complemented this analysis with an additional paragraph and a new Fig. 2, which allowed us to better qualify the differences between the products. This gives us confidence in our assessment of the product discrepancies communicated in our revised manuscript. It is worth noting, however, that while we do consider the role of uncertainty in our analysis, assessing the performance of each product in would require a full vorticity validation against *in situ* measurements, such as those measured by an array of drift boys, etc. Currently, there is not a sufficient collection of *in situ* measurements to do this analysis, and so the revised manuscript just highlights the performances of each product relative to the others. We have therefore carefully revised our phasing as not to imply that any product is more correct than another.**

*Page 2, line 34: Isn't SIC rather defined as the proportion of ice-covered water to total area (icecovered or not).*

**Corrected. Our revised definition of SIC is "a measure of the proportion of ice-covered water to total area" (Sec. 1)**

*Page 4, line 111: "horizontal and vertical components" this refers to a 2D project map with components along the vertical and horizontal directions of the grid, but can be mis-read as the vertical (3D) component of the sea-ice drift. You mean "x and y axis of the grid".*

**Corrected. Our revised phrasing in Sec. 3 refers to the $x$ and $y$ axis of the grid, and $u$ and $v$ components of the drift vectors are referred to as zonal and meridional drift components respectively.**

*Page 4, Methodology question 1: how are missing vectors dealt with for the single-sensor products: ASCAT has many missing vectors especially at lower latitudes (outskirt of the domain), the multisensor product has many more. How is the vorticity computed in case a vector is missing?*

**At every grid point i,j, the vorticity value is computed using vector measurements at i-1, i+1, j-1 and j+1. The finite difference formulation has been added to the revised manuscript in Eq. (2). Therefore, for every missing displacement vector, no vorticity is computed for that grid point and its immediate neighbors. The computation of an ice feature's mean vorticity uses valid vorticity values within the circular domain only, and therefore the total vorticity value is simply divided by a smaller**

number should there be any missing values within the domain. The revised version has clarified in Sec. 3 that "A minimum pixel validity threshold of $T$ is applied to every subdomain $D_r$, ensuring that each classified feature has an adequate number of valid vorticity values within its circumference. Subdomains that fail to meet the minimum pixel validity threshold are ignored, thus reducing the algorithm's susceptibility to classifying small regions of intense vorticity at the ice edge or coastline as features. This process is repeated independently per product with varying $r$ (500, 450 and 400 km) and $T$ (90, 85 and 80 %) parameter values, meaning that all identified features contain 180-220 valid vorticity values within their circumference, depending on the choice of $r$ and $T$". For the ASCAT sensor, most of the features detected occur in the Weddell Sea, as this is the region in which drift estimates can consistently be retrieved. The overall number of features detected using the ASCAT product is thus expectedly lower than that of the other products because its area of ice drift estimates is smaller. To overcome these limitations, in the revised analysis we have extended the region to 50°E and redefined our period of analysis to 1st June – 31st October (previously 1st April – 30th November) because the freezing months have a larger coverage of drift estimates, and they are of a higher quality flag value.

*Page 4, Methodology question 2: did you use all the vectors, irrespective of their status_flag, or did you remove some of the more dubious flags?*

We thank the reviewer for this suggestion that made us reconsider the initial analysis and strengthen it in its present form. In our revised manuscript, we used drift estimates flagged of values 20-30 (previously we used all available drift estimates, irrespective of flag status). While we acknowledge that flags 20, 21 and 22 may be of degraded quality, this potential noise can be quantified in our uncertainty analysis. Furthermore, as mentioned above, we have decided to redefine our period of analysis to 1st June – 31st October (previously 1st April – 30th November) because the freezing months have a larger proportion of higher quality flag values.

*Page 4, Methodology question 3: Do the subdomains Dr overlap? Specify in the text. If yes, the vorticity events thus contributed several times?*

Yes, vorticity features in the subdomain can overlap in both space and time. This is because we are not attempting to count the number of vorticity events, but rather to compare the rotational energy in the ice as detected from various products. It has been specified in in Sec. 3 that "Each of these subdomains represent a vorticity feature, which can overlap in space or in time, but never in both".

*Page 4, Methodology question 4: At the beginning of section 4 you refer to the "intensity distribution ... identified by the algorithm..." but your section 3 Methodology does not clearly define an "intensity". Add this in section 3, or be more specific in section 4.*

Corrected. In our revised version, we define feature intensity as "the mean vorticity of all values contained within its circumference" in our methodology section (Sec. 3).

*Page 4, Methodology question 5: "Any subdomain with a mean vorticity of zero is ignored". It seems unlikely that the mean vorticity would return exactly 0. Is you test against 0 exactly, or within a range*

*around 0 (what range?). If exactly 0 it could be worth stating in which (frequent) conditions the vorticity is exactly 0.*

**You are correct that a mean vorticity of zero is unlikely, in fact it did not occur at all. This comment was simply to inform the reader that in case a feature was measured with zero vorticity, it was neither characterized as a cyclonic or anticyclonic feature, and thus not "weighing down" either distribution's spread. This has been rephrased in Sec. 3 as "We define the feature intensity as the mean vorticity of all values contained within its circumference, and the feature variability as the standard deviation of all vorticity values contained within its circumference. Therefore, a negative intensity feature represents a circular area of sea ice dominated by cyclonic rotation, while a positive intensity feature represents an area dominated by anticyclonic rotation."**

*Page 4, Methodology question 6: Have you looked at the intensity of "significant" cyclonic and anti-cyclonic events (intensity of events above a vorticity threshold)? It could indicate if the difference you observe build from low-signal / noise events, and if the products agree better on the major events (that are possibly more relevant to the original objective of partitioning the dynamic vs thermodynamic contributions)?*

**We followed yours and Reviewer 2's suggestion and added an analysis on the intensity of "significant features" in the revised version. A temporal analysis was done on the most intense features (here defined as the 95th percentiles) and a new section 4.2 was added. This 95th percentile was then compared against products between 2013-2020 (or 2015-2020 for the AMSR-2 product due to the availability of data). The results, further discussed in Sec. 5, revealed an abrupt increase in cyclonic drift between 2014-2017, which we potentially attributed to the increased polar storms reported in the literature during this period.**

*Page 8, line 181: Before stating that there is a "a large discrepancy between products", we need to quantify what "large" is wrt the uncertainties: is it just noise or really discrepancies (observing different signals).*

**We have rephrased this in our revised version and commented on the role of uncertainties in our analysis. As described in an earlier answer to your comments, we have found that the uncertainties of the vorticity measurements are negligible, and that these discrepancies are not a symptom of the uncertainty of drift retrieval (Sec. 4.1).**

*Page 8, line 198: Here again, the merged product is described as detecting a "disproportionaly large frequency" but what if it is the most accurate of the four, and it is the larger noise in the 3 other products that leads to an underestimation of the high intensity features?*

**We have rephased this so as not to imply that any product is more or less accurate. Our uncertainty analysis also indicates that the uncertainty in the vorticity caused by the propagation of the drift uncertainty does not play a major role in the discrepancies detected (Sec. 4.1).**

*Page 8, first lines: Here again, what is the impact of the multi-sensor product having fewer missing vectors than the 3 single-sensor products?*

**This is what our results presented in the manuscript indicate. The reduction in gaps is likely to induce a change in the distribution of the more intense cyclonic events detected in the region. Besides the analysis of uncertainties described in the earlier points, we added a study of the variability of vorticity within the search region for each product (new Fig. 2). This allowed us to detect an increased dispersion at higher cyclone intensities in the merged product, which is larger than the signal found in the single sensors. We attribute this to the gap-filling procedure, and we have briefly discussed it in Sec. 5. There is indeed an impact, but currently, in the absence of independent observations that would corroborate our findings, we are unable to fully identify whether this is an artifact or a feature. The revised discussion comments on this further.**

*Page 3: line 71: missing "is" (it is necessary).*

**This has been corrected.**

*Page 3: line 74: consider having "detection" before "quantification" (1st detect, then quantify?)*

**This has been corrected. The sentence now reads "… detection and quantification of…".**

*Page 3, line 78: maybe "shared" or "common" would be better than "unique"?*

**This has been corrected. The sentence now reads "... which undergo shared processing chains…".**

*Page 3, line 85: "range" → "family"*

**This has been corrected. The sentence now reads "... SSMI/S product family…".**

*Page 3: line 93: The multi-sensor merged product does more than "treats missing data...", suggest replacing "treats missing data by means of a two-step" with "implements a two-step"*

**This has been corrected. The sentence now reads "... implements a two-step process.".**

*Page 4, line 113: "dx and dy are the grid spacing (62.5km)".*

**This has been corrected. The sentence now reads "... and $\Delta x$ and $\Delta y$ is the length and width of a pixel, which in our case are both 62.5 km".**

*Page 5 Figure 1: suggest to write the product name (ASCAT, AMSR, etc…) in the plot area in addition to of (a), (b), etc…*

**Product labels have been added to all figures to aid the inter-product comparisons. The revised figures now combine the cyclonic and anticyclonic results, resulting in an 8-panel figure, where each row separates products and each column separates cyclonic and anticyclonic features. This has been done to the newly added uncertainty (Fig. 1) and variability (Fig. 2) components, and the revised interannual variability analysis (Fig. 4)**

*Page 6 Figure 2: same suggestion as for Figure 1.*

**Same as mentioned previously.**

**Response to Reviewer 2**

Dear Dr Valentin Ludwig

Firstly, I would like to thank you for the time you have given to provide us with constructive suggestions to improve the quality of our paper. Your efforts are very much appreciated.

In this document we have provided an overview of the major changes we have made for our revised manuscript after acting upon the suggestions made by you and Reviewer 1. We have also responded to each of your specific comments, and those of Reviewer 1, in a point-by-point format. Your original comment appears in *italics*, while our response appears in **bold**.

**Responses to your specific comments:**

*Content*

*The authors de Jager and Vichi present a brief study on the intercomparison of four sea-ice drift datasets, namely the OSI-SAF merged drift product OSI-405-c and its three constituents. These are drift products derived from AMSR2, ASCAT and SSMI/S. The study focuses on the detection of cyclonic and anticyclonic rotation in the Atlantic sector of the Antarctic pack ice zone. For the years 2015-2020, the most intense cyclonic and anticyclonic features of each 48-hour period are compared statistically. The authors find that there is stronger cyclonic than anticyclonic vorticity. Comparing the products, the authors report that the merged product shows more high cyclonic vorticity values than each of the input products. This is interpreted by the authors as additional energy introduced by the merging scheme.*

*General comments*

*The paper is well-written and mostly easy to comprehend. The results are enough, both concerning relevance and quantity, to warrant publication as brief communication. My main point of criticism is that the paper is rather short on the discussion of the results. The finding that the merged product shows more high-intensity features than the singlesensor products is really interesting, but the authors do not give a reason. I understand that an in-depth analysis is beyond the scope of a brief communication, but I would like to see which ideas the authors have for further research, so that a follow-up study could build upon their work. The following questions came to my mind:*

*What is the reason for the above-mentioned mismatch between the merged product and the single-sensor products?*

*How do you judge this mismatch? Is it an artifact arising from the merging method or does it bring additional insight which the single-sensor products can not provide?*

*I would ask the authors to elaborate on this in their Discussion section. Also, it would be good to know if similar results have been achieved by other studies.*

**As mentioned in the above overview, a fundamental component of our revised version is the addition of an uncertainty and spatial variability analysis (Sec. 4.1). The results of these components allow us to conclude that this mismatch is between products is not a consequence of the uncertainty of the**

drift retrieval. We can also conclude that the spatial variability of vorticity estimates is directly proportional to the intensification of cyclonic rotation. This suggests that in regions of intense cyclonic ice drift, either:

1. sub-synoptic scale ice dynamics result in a patchy, variable vorticity field at the synoptic scale, and these are the dynamics being captured by the ice drift products. Or
2. the feature-tracking method of drift retrieval performs poorly under conditions of rapid dynamic and thermodynamic changes in ice properties which is a common phenomenon in ice under polar storms.

Our revised version shows that the variability increases more relative to the feature's cyclonic intensity for the merged product than any other product (Sec. 4.1), suggesting that the higher variability detected in the merged product is linked to a larger intensity estimate. Our analysis, however, does not allow us to discern whether the merging process results in the addition cyclonic energy in the sea-ice field, or whether the better-quality coverage of the merged product means we can more accurately extract the true variability and intensity of these features. Without a full validation of the vorticity metric – which is challenging with so such few *in situ* drift measurements – we can only hypothesize which product is performing better. We have clarified in our Discussion and Conclusions (Sec. 5). According to our knowledge, this is the first study that used multiple products to quantify rotational features in sea ice. Other studies focused on the use of individual products to determine changes in sea ice dynamics over time (Holland and Kwok, 2012; Kwok et al., 2017).This motivated our work to initially compare the products for a further quantification of the longer-term trends.

*Specific comments*

*3.1 Abstract*

*L11: Concerning the word "alternative": A bit misleading because it sounds as if the concept of sea-ice extent (SIE) would be used to quantify changes in sea-ice dynamics. I suggest to leave out "alternative".*

We have removed the word "alternative" from our abstract.

*L18: For me, the processing chain is merely the technical implementation of the merging method, therefore I would suggest to refer to the merging method here instead of the processing chain.*

This comment is no longer included in the abstract, but we have rephased the description of the 'processing chain' to "merging process" in the Discussion and Conclusions (Sec. 5).

*L18/19: I suggest to add that only cyclonic momentum is added.*

In our revised abstract, we have removed the comment suggesting that momentum is added in the merging process. Instead, we have reported that that the cyclonic features detected by the merged product are of a higher intensity and spatial variability. In our discussion (Sec. 5), we have clarified

**that although we can report on the differences between products, we cannot imply which product is more accurate without a validation of the vorticity metric.**

*3.2 Introduction*

*General: Should mention that, unlike in the Arctic, SIE in the Antarctic was quite constant until recently*

**In our revised Introduction (Sec. 1), we have added that "Antarctic SIE trends have historically been relatively constant, however, are recently characterized by pronounced variability…"**

*General: When describing your motivation, you might also want to mention more explicitly that we expect increased sea-ice drift in future, given the thinning of sea ice and the increased storminess.*

***We have added to our revised Introduction (Sec. 1) that "…this phenomenon is likely to grow in influence as extratropical cyclones shift poleward and polar storms intensify (Chang, 2017; Tamarin-Brodsky and Kaspi, 2017)"***

*L29: Do "scarce" and "sparse" not effectively mean the same thing?*

**Simplified to "sparse" only in Sec. 1.**

*L36: I would suggest to replace "ice edge" by "marginal ice zone", there is seldom a sharp and abrupt transition between sea ice and ocean which would justify the term "edge"*

**Removed and simplified to "…used to estimate the SIE…"**

*L38: Why is the variability dramatic? I would suggest something more objective and less drastic like "high" or "pronounced"*

**Rephased to "pronounced variability"**

*L39-45: Talking about limitations of SIE, you might also want to refer to Notz (2014) L56-58: I suggest to restructure and split the sentence: "Ice movement is primarily driven by . . . . Other factors are waves, ocean tilt. . . ".*

**Added comment to our Introduction (Sec. 1): "Modelled attempts to simulate the sea-ice extent have shown that uncertainties in this internal variability introduces a far greater bias than that of the satellite retrieval process (Notz, 2014)". Sentence has been split as suggested.**

*L68: Please provide a reference for your statement that the Southern ocean hosts some of the most energetic storms worldwide.*

**Comment removed.**

*L69: MIZ has not been defined yet.*

**Definition of MIZ added into the Introduction (Sec. 1) before the acronym is used.**

*L71: Much has changed since 2003/2004, please provide more up-to-date references. Also, "it therefore" should be "it is therefore".*

**Updated references to (Chang, 2017; Tamarin-Brodsky and Kaspi, 2017) in Sec. 1.**

*L74: Instead of "daily timescales", you could be more specific and speak of "two-daily resolution"*

**Comment rephrased to "48 h" in the Introduction (Sec. 1) and Discussion (Sec. 5).**

*L74: What exactly is the method which you propose? Taking the maxima and minima of the vorticity within the domain as described in L114-L120? Woud be good to state this more clearly, to me it was not immediately clear although it was the initial motivation for your paper.*

**We have clarified our methodology in Sec. 3 that "the algorithm generates virtual circular subdomains $D_r$ of radius r centred at every grid point in our vorticity field. Each of these subdomains represent a vorticity feature, which can overlap in space or in time, but never in both. We define the feature intensity as the mean vorticity of all values contained within its circumference, and the feature variability as the standard deviation of all vorticity values contained within its circumference. Therefore, a negative intensity feature represents a circular area of sea ice dominated by cyclonic rotation, while a positive intensity feature represents an area dominated by anticyclonic rotation. A minimum pixel validity threshold of $T$ is applied to every subdomain $D_r$, ensuring that each classified feature has an adequate number of valid vorticity values within its circumference.".**

*L78: As outlined above, I doubt that the term "processing chain" is appropriate here and suggest to replace it by "merging method" or something similar*

**This comment has been removed from the revised Introduction, but we have rephrased "procession chain" to "merging process" in the revised Discussion and Conclusions (Sec. 5).**

*L82-83: Isn't your conclusion that the merging introduces additional cyclonic rotation? In this case, you can also write this here instead of using the weaker formulation ". . . can induce additional. . . ". Also, you could specify already here that the additional rotational energy comes from cyclonic rotation.*

**We have rephrased the final paragraph of the introduction (Sec. 1) to better align with the results determined in our revised version. The differences we observed in the first submission have partly reduced when considering only the high-quality data, as suggested by Reviewer 1. As explained in**

the preamble, we also performed a more quantitative analysis of the uncertainties, which strengthened our results and allowed to perform a comparison of the products over the years. The revised results show that the uncertainty of the vorticity metric is negligible when using a coarse resolution drift product, and that there exists a discrepancy between the cyclonic drift detected by the various products. The products however agree in detecting an increase in the intensity of cyclonic rotation in sea ice since 2014-15 (excluding AMSR2, which is only available after this period). We focus on presenting our methodology as a useful technique for future climate index studies, and comment on the necessity to conduct a validation of the vorticity metric before we confidently assess the discrepancies between products.

*3.3 Data*

*General: Please provide references for the single drift products. Also, it would be good if you can state here which region and months you use.*

**Our data availability section has been updated with a link to an FTP server in which a user can access all the EUMETSAT OSI-SAF Low Resolution Drift products used in our revised version.**

*L90: What is meant by "SSMI/S instrument range"? Please specify.*

**For this analysis, motion vectors derived from the SSMI/S instrument family are grouped to provide a continuous dataset of measurements since 2013. This group will be analyzed as one product, termed the SSMI/S product. This has been clarified in the revised Data section (Sec. 2) as "… motion vectors derived from the SSMI/S instrument family are grouped to provide a continuous dataset of measurements since 2013. This group will be analyzed as a single product and referred to as the SSMI/S product."**

*L94: weighted by what?*

**The weighting of each single-sensor data is inversely proportional to the error of that product as shown by its validation. Therefore, the lower the uncertainty of the single-sensor product, the higher its weighting in the computation of the merged product (Lavergne et al., 2010). This has been clarified to "…the weighting of each single-sensor product is inversely proportional to the validated error of that product" in the revised Data section (Sec. 2).**

*L97: I think "coarse" would be more appropriate than "large" when speaking of resolution.*

**Rephrased from "large" to "coarse".**

*L97: Can you comment on the typical size of the cyclones which you detect in relation to the grid spacing of 62.5 km? Be careful to not mix up grid spacing and resolution.*

**The rotational features found on sea ice may originate from both oceanic and atmospheric drivers, and possibly a combination of both. Less is known about the role of mesoscale and sub-mesoscale**

oceanic processes in winter, although some initial studies indicate that sub-mesoscale processes may be relevant under ice (Biddle and Swart, 2020). There is larger evidence of the role played by atmospheric cyclones in driving sea-ice motion (Vichi et al., 2019). These large-scale synoptic features are of the order of 1000 km. Our analysis is therefore oriented towards capturing these kinds of events, assuming that sea ice would be affected at the same scale of the synoptic events. The analysis was done by forcing the search domain as a circle, and we tested the sensitivity to 400/450/500 km radius. The domain included approximately 150-210 grid points. The product grid size is therefore sufficient to resolve the features of interest. We have added this consideration in Sec. 5 as follows: "…our detection algorithm identifies circular ice drift features with a radius of 450 km ± 50 km – which is about 6-7 times the spatial resolution of the products and of the scale of atmospheric weather – and quantifies the characteristics of the vorticity field within this circumference."

*L101: Please specify the projection (NSIDC projection with the latitude of true scale at 70∘S?) or give a reference.*

**Antarctic OSI-405-c motion vectors are mapped onto a NSIDC polar stereographic projection (latitude of true scale at 70°S). This has been added in the Data section of the revised manuscript.**

*3.4 Methodology*

*L105: The readability here and in the rest of the paper would be better if you could adopt the practice to use "sea-ice" when speaking about sea-ice properties (sea-ice vorticity, sea-ice concentration etc) and use "sea ice" when referring to it as a noun.*

**In the revised version, we hyphenate the term when speaking of related properties, and not when using it as a noun. This has been done throughout the manuscript.**

*L107-110: Domain and months should be specified in the Data Section. It would also be good if you could state how large the area is in km². What is your criterion for the "ice-covered area"? SIC above 15 %? Please state this here.*

**In our revised version, we have included the spatial and temporal ranges in the Data section (Sec. 2) as follows: "Due to the limitations of measuring sea-ice drift in melting conditions and during periods of insufficient ice cover, only the months of June-October were considered, and our analysis focused on the Atlantic Sector of the Southern Ocean, spanning the area between 65° W and 50° E.". The initial manuscript showed results between 1st April – 31st November. The revised manuscript shows results between 1st June – 31st October. This change was made because the freezing months had a larger area of ice coverage with fewer rejection quality flags. The initial manuscript showed results between 65° W and 10° E. The revised manuscript shows results between 65° W and 50° E. This change was made to reduce the influence of the region boundary 10° E, an area where sea-ice coverage band is wide. The sea-ice coverage at 50° E is thin relative to the scale of features being detected, and so the algorithm is not affected by an artificial boundary parameter. The total area of a feature is approximately $6.36\times10^5$ –$5.72\times10^5$ km² depending on the choice of free parameters**

*L115: If you choose the maxima/minima of the mean vorticities, you might get into trouble if there are outliers which are not representative of typical cyclonic/anticyclonic features. Can you comment on this? Did you compare the results which you get by taking the extreme values to the results which you would get if using a more robust estimator like the 95th percentile? Would you expect differences arising from this? Please briefly discuss.*

**As mentioned earlier, we improved our methods section explaining how features are identified and categorized. This now reads as:**

**"…the algorithm generates virtual circular subdomains $D_r$ of radius $r$ centred at every grid point in our vorticity field. Each of these subdomains represent a vorticity feature, which can overlap in space or in time, but never in both. We define the feature intensity as the mean vorticity of all values contained within its circumference, and the feature variability as the standard deviation of all vorticity values contained within its circumference. Therefore, a negative intensity feature represents a circular area of sea ice dominated by cyclonic rotation, while a positive intensity feature represents an area dominated by anticyclonic rotation. A minimum pixel validity threshold of $T$ is applied to every subdomain $D_r$, ensuring that each classified feature has an adequate number of valid vorticity values within its circumference. Subdomains that fail to meet the minimum pixel validity threshold are ignored, thus reducing the algorithm's susceptibility to classifying small regions of intense vorticity at the ice edge or coastline as features".**

**Regarding your comment on outliers, we agree that erroneously high vorticity pixels will impact the mean vorticity within the search domain. Currently the influence of these high pixels is reduced by the minimum valid pixel requirement, meaning any detected feature will have at least 80/85/90% non-missing vorticity values, which is approximately 190 valid vorticity values for each feature. As indicated in an earlier answer, in our revised version we have only used quality of measurements according to the status flag information provided in the dataset, as well as propagate the uncertainty measurements into our vorticity computation. This allowed us to quantify the confidence of our vorticity measurements. An analysis was done on 'significant' features (i.e., the 95th percentile) as suggested by yourself and Reviewer 1.**

*3.5 Results*

*General: Please state how many data points there were per year. Was it always the same number?*

**We now include the number of features detected in our revised Results section (Sec. 4.1). "A total of 311 758 anticyclonic features were detected over this period, with the most being returned by the merged product (144 491), followed by the AMSR-2 (84 171), SSMI/S (64 263) and ASCAT (18 383) products. The number of cyclonic features returned was similar in the single-sensor AMSR-2 (105 889), SSMI/S (66 111) and ASCAT (21 991) products, while the merged product detected approximately twice as many cyclonic features (295 484) as anticyclonic features. A total of 489 475 cyclonic features were returned by all four products, 57 % more relative to the number of anticyclonic features detected."**

*Figure 1: If your main goal is to compare the products among each other, it might make more sense to have one panel per year instead of one panel per product. If it does not overload the plots, you could also*

*consider merging the panels a–d to one and mark the products by different colors. With the current alignment, I find it hard to compare the results of single years between sensors. Same for Fig. 2.*

**We have now made a single figure of 8 panels (4 rows; 2 columns) for Figure 1, 2 and 4 of our revised version, where each product results can be compared between rows, and cyclonic and anticyclonic results can be compared between columns.**

*L124-126: This technical description of the box-and-whisker plot could be moved to the caption of the Figure.*

**Figures 1 and 2 of the initial manuscript showed that interannual distribution of all cyclonic and anticyclonic features respectively. Following yours and Reviewer 1's suggestion to consider 'significant' features, Figure 4 in Results (Sec. 4.2) now shows the interannual distribution of the 95$^{th}$ percentile of features. Both cyclonic and anticyclonic features are shown on the same figure to aid the visual comparison, as this was another suggestion made by yourself. We have now moved the technical description of the box-and-whisker plot to the caption of Figure 4. Figures 1 and 2 now show the results of the uncertainty and variability analysis respectively, which are two components added to the revised manuscript as mentioned in the preamble.**

*L131: This is the kind of statement which is hard for me to assess if the four ice motion products are shown in separate panels. Can you give the values to which you refer here in a Table?*

**As mentioned above Figures 1 and 2 of the initial manuscript showed that interannual distribution of all cyclonic and anticyclonic features respectively, and following yours and Reviewer 1's suggestion to consider 'significant' features, Figure 4 in Results (Sec. 4.2) now shows the interannual distribution of the 95$^{th}$ percentile of features. Both cyclonic and anticyclonic features are shown on the same figure to aid the visual comparison. This means that Figure 4 has 8 panels (4 rows; 2 columns), where each product results can be compared between rows, and cyclonic and anticyclonic results can be compared between columns. Furthermore, we have now added Table 2 in the revised Results (Sec. 4.1) which shows the statistical mean and standard deviation of the data spread to supplement the communication the interannual variability analysis shown in the revised Figure 4.**

*L132: What exactly do you refer to by "spread"? Interquartile range? Range between whiskers? Further, it would be good to also at least mention the actual magnitude of the cyclonic features, not only the spread, even if the latter is your main focus.*

**In our revised Results section (Sec. 4.2), we have quantified the interannual intensity distribution by the mean and standard deviation of the 95$^{th}$ percentile of cyclonic and anticyclonic features, as these two metrics indicate both the magnitude of intensity and the 'spread' of the distribution. These metrics have been reported in the text (Sec. 4.1) and in Table 2.**

*L139: What do you mean by "high levels of interannual variability"? I do not have the impression that the medians or interquartile ranges in Fig. 1b vary more than in a, c or d.*

We have modified our interannual variability component of our analysis in our revised version. Following suggestions made by yourself and Reviewer 1, we now present the interannual variability of the 95th percentile of features in Section 4.2. This distribution is illustrated in the new Figure 4. Furthermore, we have quantified the distribution as the mean ± standard deviation of the 95th percentile of features. These values have also been reported in Table 2, which was another addition to the revised manuscript following your suggestion.

*L141: "which being detected": Should this be "which was detected"?*

This comment has been corrected as suggested.

*L144: Should mention the reduced y-axis range of Fig. 2 compared to Fig. 1.*

As mentioned above Figures 1 and 2 of the initial manuscript showed that interannual distribution of all cyclonic and anticyclonic features respectively, and following yours and Reviewer 1's suggestion to consider 'significant' features, Figure 4 in Results (Sec. 4.2) now shows the interannual distribution of the 95th percentile of features. Both cyclonic and anticyclonic features are shown on the same figure to aid the visual comparison. We have also scaled the y-axis to the same range on all subplots in Fig. 4 to aid comparison.

*L170-176: Please put the IQR and σ values in a Table, this would be much easier to grasp and would improve the readability.*

We have included Table 2 in our revised Results (Sec. 4.2), which includes the mean, standard deviation and 95th percentile threshold for each year to supplement the intensity distribution illustrated in the new Figure 4. The mean ± standard deviation of the uncertainty and spatial variability analysis components in Section 4.1 have also been reported in a table format (Table 1).

*3.6 Discussion and Conclusions*

*L187: ". . . increasing trend. . . ": Do you refer to the spread or to the absolute values of vorticity?*

In Section 4.2 of our revised Results, we consider the mean ± standard deviation of the 95th percentile of features. The interannual distribution of these features is illustrated with box-and-whisker plots in revised Figure 4, while the mean, standard deviation and 95th percentile thresholds are reported in Table 2. We have clarified our comments that "there is no obvious interannual trend shown in anticyclonic intensity between 2013–2020 (Fig. 4a, c, e and g), visually indicated by the box-and-whisker rectangles remaining relatively constant in all four products and with few outlier features detected. This was further supported by the relatively small variability in the mean intensity and standard deviation between years (Table 2). Conversely, a clear cyclonic interannual trend was evident, characterized by relatively low-intensity features in 2013 and 2014, followed by an abrupt increase in intensity from 2015–2017 (Fig. 4b, d, f and h). From 2014–2017, the mean of the most intense cyclones increased by a factor of 2.8, 2.6 and 1.7 for the ASCAT, SSMI/S and merged products respectively (Table 2). Much like the variability-intensity cyclonic trends described earlier (Fig. 2b,

**2d, 2f and 2h), it was again evident that the standard deviation increases with the intensity from 2014–2017 for each available product (Table 2).”**

*L187: Please discuss the robustness of this trend, given that your study period is quite short. Is this also found by other studies?*

**In our revised version, we have made two changes which improve the robustness of our analysis. Firstly, we have incorporated an uncertainty analysis into our analysis as suggested by Reviewer 1 (Sec. 3 and Sec 4.1). We theoretically propagated the data error onto the vorticity estimates according to Dierking et al. (2020). This gives us confidence that the discussed trends are not caused by noise in the drift retrieval estimation. Secondly, we have modified our trend analysis to consider the 95th percentile of each year (please see more details in the preamble). This resulted in a new section 4.2 added to the revised version. This has allowed us to see a clear change in the distribution of cyclonic drift, particular from 2014-2017. According to our knowledge, this is the first study that used multiple products to quantify rotational features in sea ice. Other studies focused on the use of individual products to determine changes in sea ice dynamics over time (Holland and Kwok, 2012; Kwok et al., 2017).This motivated our work to initially compare the products for a further quantification of the longer-term trends. There is, however, evidence that an increase in the influence of polar storms between 2014-2017 primarily drove the rapid decrease in SIE over the same time period (Wang et al., 2019), which coincide with the increase cyclonic drift reported in our revised version. These considerations have been added in the revised Discussion in section 5.**

*L197-202: Very interesting indeed to see that the merged product shows more vorticity than any of the others. I would like to see this discussed in more detail. An in-depth discussion would probably be too much, but can you elaborate on potential reasons or give directions for future research? Also, do you trust this result? Would be good to get an idea whether the additionally introduced rotation is valuable information which we can not get from the single-sensor observations or whether it is an artifact of the merging.*

**As explained in the preamble, we have expanded our analysis beyond a brief communication at the suggestion of the handling editor. For this reason, we include a more in-depth analysis of the product discrepancies, separating our revised Result section 4 into two subsections. Sec 4.1 is dedicated to the analysis of uncertainties and the comparison between products, which is an improved version of our earlier brief communication. Here we rephrase our results as not to imply any product is more correct than another, as this would require a full validation of the vorticity metric to confirm. Instead, we highlight likely – or less likely - drivers of this discrepancy. Firstly, we show that the uncertainty is unlikely to cause the difference between products. Secondly, we report a much higher variability in the spatial vorticity field when detecting cyclonic features compared to that of anticyclonic features. We also show that this variability increases with higher intensity features, and that the merged product detected the highest intensity and most variable features. We offer an interpretation of these results in the revised Section 5: “Our analysis, however, does not allow us to discern whether the merging process causes an amplification of cyclonic drift in the sea-ice field, or whether the better coverage of the merged product means it can more accurately detect the rotational drift in the ice compared to the single-sensor products. We hypothesize that the large variability introduced by the merging process is causing an artificial intensification of cyclonic rotation. However, in the absence of independent observation that would corroborate our findings, we are unable to fully identify whether this is an artefact or a feature”.**

*L204-205: Please give a reference or explain why you expect disproportionately high frequency of low-intensity features in the Eastern Weddell Sea.*

**This comment referred to a spatial analysis done using the most intense features detected by each of the four products (not shown in the original manuscript), which has not been included or commented on in the revised version. This was expected due to the patterns in coverage of the ASCAT product. More specifically, the ASCAT product usually has the lowest coverage our of any of the products, and the Eastern Weddell Sea is the region in which the ASCAT consistently had valid drift estimates, compared to the poorer coverage closer to the MIZ. Furthermore, the vorticity field in the Eastern Weddell Sea was typically of a lower intensity than that closer to the MIZ.** We decided to redefine our period of analysis to between 1st June – 31st October (previously was 1st April – 30th November) to improve the quality and coverage of single-sensor drift estimates used in this study. This was decided because the freezing months have a better coverage of drift estimates. Furthermore, **the initial manuscript showed results between 65° W and 10° E, but the revised manuscript shows results between 65° W and 50° E. This change was made to reduce the influence of the region boundary 10° E, an area where sea-ice coverage band is wide. The sea-ice coverage at 50° E is thin relative to the scale of features, and so no features are detected here because of the insufficient ice cover, and not by an artificial boundary parameter. This increased the available sea-ice area for all products on which the algorithm could identify rotational features**

*L209: See my comment to your L74. Please describe the method briefly, since it was the main motivation for your work.*

**We have clarified our methodology in Section 3 that "the algorithm generates virtual circular subdomains $D_r$ of radius r centred at every grid point in our vorticity field. Each of these subdomains represent a vorticity feature, which can overlap in space or in time, but never in both. We define the feature intensity as the mean vorticity of all values contained within its circumference, and the feature variability as the standard deviation of all vorticity values contained within its circumference. Therefore, a negative intensity feature represents a circular area of sea ice dominated by cyclonic rotation, while a positive intensity feature represents an area dominated by anticyclonic rotation. A minimum pixel validity threshold of $T$ is applied to every subdomain $D_r$, ensuring that each classified feature has an adequate number of valid vorticity values within its circumference.".**

*L212-219: Please give directions/ideas how to find out the reason for the mismatch in the cyclonic drift features.*

**In the revised Discussion section (Sec. 5), we have proposed a potential reason for the product disagreement as suggest by yourself, while also not implying that any product is more accurate than another until a validation of the vorticity metric is done, as this was a concern raised by Reviewer 1. We comment: "Our analysis, however, does not allow us to discern whether the merging process causes an amplification of cyclonic drift in the sea-ice field, or whether the better coverage of the merged product means it can more accurately detect the rotational drift in the ice compared to the single-sensor products. We hypothesize that the large variability introduced by the merging process is causing an artificial intensification of cyclonic rotation. However, in the absence of independent**

observation that would corroborate our findings, we are unable to fully identify whether this is an artefact or a feature. Furthermore, the feature-tracking method of drift retrieval may be susceptible to error under conditions of rapid dynamic and thermodynamic changes in sea-ice properties, such as in the event of a strong cyclone traversing the sea ice. It is therefore necessary to consider that rapidly moving ice floes may be blurring the rotational drift we are attempting to estimate over a 48 h period".


**Overview of our revised manuscript after major changes:**

Following comments received from the reviewers, we have decided to add a further subsection to our revised Results section, which now includes two subsubsections each with a new figure. Reviewer 2 requested to add some examples of the vorticity fields that are used by the detection algorithm. We have therefore provided two case studies: Case Study 1 (Sect. 4.1.1) which considers the effect of an atmospheric cyclone on the underlying sea ice dynamics (illustrated by the new Fig. 1), and Case Study 2 (Sect. 4.1.2) which considers the effect of a persistent high-pressure cell on the underlying sea ice dynamics (illustrated by the new Fig. 2). Both conditions were observed during expeditions of the SA Agulhas 2 in the region. This addition also allowed us to expand on a few considerations regarding the comparison between the products and their relative quality. The previous Sec. 4.1 is now Sec. 4.2, which also contains references to the case studies to highlight how they compare with the bulk of the rotational features detected by the algorithm. The contents of the previous Sect. 4.1 and Sect. 4.2 are now found in Sect. 4.2 and Sect. 4.3 respectively, along with their corresponding figures, now labelled Fig. 3 and 4.

The other major change is made following comments by Reviewer 1, who suggested we convert the scatterplots of the previous Fig 1. and Fig. 2 into 2-D histograms. We agree with Reviewer 1 that this would improve the readability of our figures, and these modified figures are now found in the revised Results Sect. 4.2 labeled as Fig. 3 and Fig. 4 respectively. The 2-D histograms now show the clustering of data points around the regression line more clearly, which aids the visualization of the discrepancy in the number of features detected by each product with the colormap scheme. The values of the slope, intercept, standard error of the slope and the number of features detected are now also shown on the figure panels, as suggested by both Reviewer 1 and 2.

**Response to Reviewer 1**

Dear Dr Valentin Ludwig

Once again, I would like to thank you for the time you have given to provide us with constructive suggestions to improve the quality of our paper after a second round of review. Your efforts are very much appreciated.

In this document we have provided an overview of the major changes we have made for our revised manuscript after acting upon the suggestions made by you and Reviewer 2. We have also responded to each of your specific comments, and those of Reviewer 2, in a point-by-point format. Your original comment appears in *italics*, while our response appears in **bold**.

**1. Content**

*The authors de Jager and Vichi re-submitted a paper which analyses the differences between different satellite-based sea-ice drift products from EUMETSAT OSI-SAF (product OSI-SAF 405 c), namely five single-sensor-based products (AMSR2, ASCAT, 3x SSMIx) and a merged product based on the formerly mentioned ones. Much content has been added to expand the manuscript from the previously submitted "Brief communication" to a full-fledged research article. The major differences are the inclusion of an uncertainty estimate and analysis, the discussion of spatial variability and the restriction of the analysis to high-quality data as per the quality flags of the drift product.*

**2. General Comments**

*I appreciate the effort which the authors made to submit this expanded version. I fully agree with their decision to extend the manuscript to a full research article, following the handling editor's recommendation. I enjoyed reading this manuscript and found my concerns from the first round of review addressed satisfactorily. However, I still have some comments which I would like to see addressed before publication. Line numbers in my specific comments correspond to the track-change version of the revised manuscript.*

**3. Specific Comments**

**3.1 Abstract**

*L14: " . . . Antarctic ice characteristics.": You might consider removing "Antarctic", since the need for methods to quantify sea-ice changes is also there in the Arctic.*

**The adjective "Antarctic" has been removed, and the sentence now reads: "…develop methods to quantify changes in sea-ice dynamics that would indicate trends in the ice characteristics."**

**3.2 Introduction**

*General: I find the Introduction quite long. The part after L72 was interesting and well written, but the part before could for my taste be condensed. I suggest to build your 1 motivation on the increased storminess and the recent changes in Antarctic sea ice, but not so much on having an additional measure for assessing climatological trends as a complement to SIE.*

**We thank the reviewer for their comment and agree that our motivation and introduction text could be improved to better introduce the reader to the necessary concepts, but without detracting from the main points being presented. We have therefore removed some unnecessary text from the introduction – and rearranged other portions – to streamline the ideas into a more coherent flow. As your comment suggests, we build our motivation around the increased storminess and variability since 2014, which then instinctively proposes the need to detect and quantify the effect of atmospheric weather on sea ice and apply those considerations onto the existing datasets. We feel that the revised introduction better prepares the reader to understand the usefulness of our results, but without deviating too much from the main messages communicated in the Discussion.**

*L44-45: A reference would be good, for example IPCC AR6.*

**The IPCC AR6 2021 report has been referenced in text as suggested. The sentence now reads: "…presented to highlight the effects of global warming (Masson-Delmotte et al., 2021)." The corresponding citation has also been added to the references section:**

**Masson-Delmotte, V., Zhai, P., Pirani, A., Connors, S. L., Péan, C., Berger, S., Caud, N., Chen, Y., Goldfarb, L., Gomis, M. I., Huang, M., Leitzell, K., Lonnoy, E., Matthews, J. B. R., Maycock, T. K., Waterfield, T., Yelekçi, O., Yu, R. and B., Z.: Climate Change 2021: The Physical Science Basis. Contribution of Working Group I to the Sixth Assessment Report of the Intergovernmental Panel on Climate Change., 2021.**

*L72: You start your sentence with "Studies. . . ", but then refer to only one. I suggest to either rephrase or add more references.*

**We have now added a second reference (Kwok et al., 2017) with the corresponding citation in the references section:**

**Kwok, R., Pang, S. S. and Kacimi, S.: Sea ice drift in the Southern Ocean: Regional patterns, variability, and trends, edited by J. W. Deming and E. C. Carmack, Elem. Sci. Anthr., 5, doi:10.1525/elementa.226, 2017.**

*L77: "El Nino" → "El Niño"*

**The eñe character has been corrected in the revised version.**

*L85: Replace "ice floes" by "ice pack"*

**The word "floes" has been replaced with "pack", and the sentence now reads: "…as strong winds induce synoptic scale rotation into the ice pack while carrying…"**

*L96: I suggest to use present tense throughout the paper whenever possible, especially when describing your own work.*

**In our revised version we have now used present tense throughout.**

*L102: Please mention that all products are from OSI-SAF.*

**The sentence now reads: "…by computing the sea-ice vorticity using satellite ice drift estimates from EUMETSAT OSI SAF and quantifying the rotational drift of the sea ice within circular domains."**

*L105: "assess" → "to assess"*

**This sentence has been corrected and now reads: "…and to assess to what extent ice drift product…"**

*L106: Please specify the period by saying which years you refer to.*

**The specific temporal range has been included and the sentence now reads: "…to capture changes over the period of data availability (2013-2020)."**

3.3 Data

*Generally: Please spell out the acronyms when you use them for the first time.*

**In the revised version, we have expanded the following acronyms in sections:**

**1 Introduction:**

 **"EUMETSAT" now reads "European Organization for the Exploitation of Meteorological Satellites (EUMETSAT)…"**

 **"OSI SAF" now reads "Ocean and Sea Ice Satellite Application Facility (OSI SAF) …"**

**2 Data:**

 **"AMSR-2" now reads "…Advanced Microwave Scanning Radiometer 2 (AMSR-2) …"**

 **"ASCAT" now reads "…Advanced Scatterometer (ASCAT) …"**

 **"SSM/I" now reads "…Special Sensor Microwave Imager (SSM/I) …"**

**"SSMI/S"** now reads **"…Special Sensor Microwave Imager/Sounder (SSMIS) …"**

**"JAXA"** now reads **"…Japan Aerospace Exploration Agency (JAXA) …"**

**"DMSP"** now reads **"…Defense Meteorological Satellite Program (DMSP) …"**

*L123: Is it intentional that you write "SSMIS-f17" or should it be SSMIS-F17 in the following lines)?*

**No, this detail was overlooked. In our revised version, we have capitalized the "F" in the DMSP platform names (Sect. 2 and Sect. 4.2).**

*L125:"SSMI/S" → "SSMIS"*

**This sentence has been rephrased and now reads "… motion vectors derived from the SSM/I and SSMIS instruments are grouped to provide a continuous dataset of measurement from 2013. This group will be analyzed as a single product and referred to as the SSMI/S product."**

*L136: Please provide a reference (URL) for the NSIDC projection.*

**A URL reference for the polar stereographic projection has been included in the revised version. The sentence now reads "Antarctic OSI-405-c motion vectors are mapped onto a NSIDC polar stereographic projection (https://nsidc.org/data/polar-stereo/ps_grids.html) and are available …"**

*L139: A reference for the limitations would be nice*

**Two citations have been added, namely Lavergne (2016) and Sumata et al. (2015), with the following text added to the reference section:**

**Lavergne, T.: Algorithm Theoretical Basis Document ( ATBD ) for the OSI SAF Low Resolution Ocean & Sea Ice SAF Algorithm Theoretical Basis Document for the OSI SAF Low Resolution Sea Ice Drift Product GBL LR SID — OSI-405-c, , (July), doi:10.13140/RG.2.1.3082.3920, 2016.**

**Sumata, H., Kwok, R., Gerdes, R., Kauker, F. and Karcher, M.: Uncertainty of Arctic summer ice drift assessed by high-resolution SAR data, J. Geophys. Res. Ocean., 120(8), 5285–5301, doi:10.1002/2015JC010810, 2015.**

3.4 Methodology

*L145: "SSMI/S" → "SSMIS". Please also check for other occurrences in the paper.*

Here the acronym "SSMI/S" is referring to the name we have termed the grouped ice drift datasets derived from the both the SSM/I and SSMIS instruments. It is not specifically referring to either the SSM/I or SSMIS products. In Sect. 2, we mention that "this group will be analyzed as a single product and referred to as the SSMI/S product."

*L152: "Acceptable" sounds quite subjective. Readers who are not familiar with the meaning of the single flags (such as myself) might wonder why you choose flag 20 as criterion. Please give very briefly some information about the meaning of the flag numbers, and your reason to choose flag 20 as threshold.*

In the revised Methodology section (Sect. 3), we have included an explanation of the product's flag values and clarified the reasons for our choice. The following text has been added: "Low-quality flagged drift estimates (i.e., flag values 0-19 in the OSI-405-c product) were rejected, while only those flagged with a good quality index (i.e., flag values 20-30) were considered. Rejected and good quality index flags are determined by the quality of the drift estimate retrieval and are made available in the OSI-405-c product. Nominal quality estimates (i.e., flag value of 30) have the lowest retrieval uncertainty and were measured independently of its neighbours, while flag values 20-29 included drift estimates that required a correction or interpolation scheme from neighbouring locations and therefore have a larger uncertainty. Rejection quality flags correspond to locations where no valid drift estimate could be made, and therefore no vorticity values are computed at those grid points or their adjacent neighbours (more information on the rejection and quality index flags can be found in Sect. 4.4 of the EUMETSAT OSI SAF Low Resolution Sea Ice Drift User Manual found at: https://osisaf-hl.met.no/sites/osisaf-hl/files/user_manuals/osisaf_cdop2_ss2_pum_sea-ice-drift-lr_v1p8.pdf). The 20-30 range of flag values was chosen for this analysis because each drift estimate has a corresponding drift uncertainty (as from 1st June 2017), and so while some non-nominal drift vectors may be of degraded quality, this potential noise was quantified and propagated through the vorticity computation."

*L159-161: I wondered if there may be cases where there is one cyclonic and one anticyclonic feature in Dr. If yes, taking their mean would erroneously return comparably small rotational activity. Can this happen? Please elaborate on this in the Discussion.*

Yes, you are correct that it is possible for one single feature (Dr) to partially capture the vorticity of both a cyclonic and anticyclonic feature. In these cases, the resultant mean vorticity would be relatively low, and therefore represented in the low-intensity portions of Fig. 3, 4 and 5. However, it is unlikely that two vorticity features at the scale of atmospheric weather would be so close to each other that the opposing rotation of both features are captured in a single 450 km radius. These cases are therefore rare and make up a very small number of the total ≈ 800,000 features detected from 2017-2020 (or 2016-2020 in the case of Fig. 5). We have included the following text in the revised Discussion (Sect. 5) to communicate this point to the reader: "It is also necessary to consider that the search radius of the feature detection algorithm described in Sect. 3 can be affected by contiguous cyclonic and anticyclonic features in the sea ice. Such condition would

**have a neutralizing effect on the value of its mean vorticity, and so these kinds of features should be represented in the low-intensity portions of Fig. 3, 4 and 5. Assuming a random distribution of these features, we expect them to be highly heterogenous; however, only the variability of anticyclonic features detected by the merged product show higher heterogeneity in the low-intensity features (Fig. 4g). None of the products report low-intensity cyclonic features with high variability (Fig 4b, 4d, 4f, and 4h). We thus conclude that close, dipole-like features in the vorticity field are relatively uncommon, or that they are spatially more extended, and it is thus unlikely that two opposing rotation features are equally captured in the same 450 km search radius.”**

*L168: 1) "resolution" should be "grid spacing". Also, I suggest to simply write ". . . a grid spacing of 62,5 km is fine enough to capture these features.", the current formulation sounds a bit odd.*

**The sentence has been rephrased and now reads: “…and therefore the 62.5 km grid spacing of the OSI-405-c product is fine enough to capture these features.”**

*L172: Dierking et al. (2020) is missing in your reference list.*

**The following missing reference has now been added to the revised reference list:**

**Dierking, W., Stern, H. L. and Hutchings, J. K.: Estimating statistical errors in retrievals of ice velocity and deformation parameters from satellite images and buoy arrays, Cryosphere, doi:10.5194/tc-14-2999-2020.**

*Equation 2: $\Delta T$ is probably constant, right? Does this mean that $\sigma_{vort}$ scales linearly with $\sigma_{tr}$? Please mention this briefly.*

**No, $\Delta T$ is not necessarily constant. The OSI-405-c product derives drift estimates from two overlapping satellite swath measurements at $t_0$ and $t_1$. These swaths, however, are not exactly 48 h apart, but instead can range from 40-56 h apart. In the case of the single-sensor drift estimates, this means that $\Delta T = 48$ h $\pm 8$ h in the Antarctic ($\pm 5$ h in the Arctic) depending on the timing of the overlapping swaths. In the case of the merged product, the product combines at least three single-sensor measurements into a single estimate, and so $t_0$ and $t_1$ are ambiguous. Measurements at $t_0$ and $t_1$ are therefore artificially set to midday (12h00 UTC) exactly 48 h apart. In the revised Methodology section (Sect. 3), we have added the following sentences: “In this case, $L$ remains constant at 62.5 km. For the single-sensor products, $\Delta T$ varies within a 48 h $\pm 8$ h range depending on the timing of the two overlapping satellite swaths, while $\Delta T$ for the merged product is artificially set to 48 h.”**

General: How is the merged product's uncertainty calculated? As a Gaussian error propagation based on the other three product's uncertainties? Please describe this.

**Yes, you are correct that the merged product's uncertainty values are computed with a Gaussian error propagation function based on the uncertainties of the single-sensor uncertainty values available at that grid point. We have clarified this in the results, which now reads: "This result is intuitive as the merged product is created using a combination of the single-sensor products and its uncertainty is computed using a Gaussian error propagation function based on the uncertainties of its constituents, and thus the resultant uncertainty of the merged product is expectedly more variable than that of the single-sensor products (Lavergne, 2016)."**

3.5 Results

*Figs 1&2: These are important Figures, but it would be really helpful to use a density plot or 2d histogram instead of a simple scatter plot. With this amount of points, it is visually impossible to see the actual distribution of values. This would also help to assess the reliability and representativeness of the trend lines. Can you provide some estimation of the robustness of the fit lines, for example a standard error of the linear regression? Also, it would be good to add the number of points, the slope and the intercept, either in the Figures themselves or in the caption.*

**We thank the reviewer on their suggestion of how to improve the readability of our figures. We have followed their advice and changed our Figures 1 and 2 in the original Results (Sect. 4.1) from scatterplots to two-dimensional histograms in the revised version. We decided to use counts instead of density to highlight the difference between the products, since the ASCAT product has much less points. Since we are interested in comparing the available products, we found that a density plot made them more homogeneous and discarded important features. Note that these figures are now labelled Fig. 3 and 4 and found in Sect. 4.2 in the revised version. These histograms required a new colormap scheme, and so the previous "Red for anticyclonic features" and "Blue for cyclonic features" scheme has been discarded. A color-bar corresponding to the new two-dimensional histogram has also been added. The slope, intercept, standard error of the slope and number of features are also now included on each panel of the revised figures. Below we show the original (left) and revised (right) variability vs intensity figure:**

[Figure]

*General: I was a bit surprised about the discrepancy between AMSR2 and SSMIS. Since they are both passive-microwave based, I would have expected them to be closer to each other than to the active-microwave based ASCAT data. Can you elaborate on this at some point?*

**The AMSR-2 and SSMI/S products are indeed the most similar, however, there is still a discrepancy in the vorticity fields computed by each product. This is partly caused by the quality of the drift vectors according to each product. As shown in the previous answer and in the revised Fig. 3 and 4, the SSMI/S product has less valid vorticity points than the AMSR-2 product. The case studies we added in the revised Sec. 4.1 are also meant to give some examples of these differences. The impact of rejection quality estimates is illustrated in the new Fig. 1 and 2, in which we demonstrate that the SSMI/S product has a slightly higher frequency of rejection-quality estimates compared to the AMSR-2. It is important to note that for every pixel that has a rejection quality drift estimate, no vorticity value can be computed for any of its adjacent pixels. This means that 1-4 pixels of the vorticity field are lost for every 1 pixel rejected, and so the differences between displacement fields appears to be exaggerated in the vorticity metric. We add the following comment into the revised discussion: "The better coverage of good quality-flagged drift estimates from the AMSR-2 product means that its resultant vorticity field has better coverage than the SSMI/S product, despite both being passive-microwave based, while the coverage of the active-microwave based ASCAT product is considerably worse than the other three products."**

*L218: What do you mean by "gradient factor"? The slope of the fit line? If yes, I would suggest to call it "slope".*

**We have rephrased "gradient factor" to "slope" throughout this portion of text in Results (Sect. 4.1). The text now reads: "The merged product had the largest slope of 0.44 for cyclonic features**

**(Fig. 2h), and the single-sensor AMSR-2 (Fig. 2b), ASCAT (Fig. 2d) and SSMI/S (Fig. 2f) products had smaller slopes of 0.31, 0.33 and 0.34 respectively. No obvious relationship between intensity and associated variability was seen for anticyclonic features, with the single-sensor AMSR-2 (Fig. 2a), ASCAT (Fig. 2c) and SSMI/S (Fig. 2e) products all showing a slope of approximately 0.01. The merged product had an inversed proportionately slope of -0.10 (Fig. 2g), indicating that intense anticyclonic features are more homogenous."**

*L237: "intensity" → "intensities"*

**The word "intensity" has been corrected to "intensities", and the sentence now reads: "Results show that the intensities of cyclonic features…".**

*L238-245: Are these the same features that went into Figures 1 and 2? If yes, you could consider to add the numbers of features either to Table 1 or to the Figure captions. Also, why does AMSR2 detect so much more features than SSMIS?*

**No, these are not the same pool of features in Fig. 1 and 2. Fig. 1 and 2 (which are now labelled Fig. 3 and 4) include all features detected from 2017-2020, as this is the period in which uncertainty values are included in the OSI-405-c drift estimates. The values mentioned in lines 238-245 (and represented in the previously labelled Fig. 3, but now labelled Fig. 5 in the revised version) are the number of features detected from 2016-2020, as this is the longest period of overlap available for all four products (no AMSR-2 derived estimates are available before September 2015).**

**As mentioned in a previous comment, the SSMI/S product has a slightly higher frequency of rejection-quality estimates compared to the AMSR-2. It is important to note that for every pixel that has a rejection quality drift estimate, no vorticity value can be computed for any of its adjacent pixels. This means that 1-4 pixels of the vorticity field are lost for every 1 pixel rejected based on the poor-quality drift estimate, and so the differences between vorticity fields appears enhanced in the vorticity metric. In the revised version, the slight difference in rejection quality estimates is illustrated in the new Fig. 1 and 2. We add the following comment into the revised discussion: "The better coverage of good quality-flagged drift estimates from the AMSR-2 product means that its resultant vorticity field has better coverage than the SSMI/S product, despite both being passive-microwave based, while the coverage of the active-microwave based ASCAT product is considerably worse than the other three products."**

*L261: ". . . than _for_ anticyclonic features."*

**This sentence has been corrected and now reads: "The 95th percentile intensity threshold was therefore 1.5-2.0 times larger for cyclonic features than for anticyclonic features."**

*L270: "into" → "of"*

**The word "into" has been replaced with "of", and the sentence now reads: "A further analysis of the interannual variability…".**

*L278: What is meant by "statistical " mean?*

**We acknowledge that the adjective "statistical" is redundant in this context and so it has been removed. The sentence now reads: "The mean, standard deviation and intensity threshold…"**

*L279: AMSR2 is in use since late 2012, it is the drift product which is available since 2015.*

**We have modified our phrasing in text to clarify that we are referring to the AMSR-2 drift product and not the AMSR-2 instrument. The sentence now reads: "Note that the AMSR-2 drift product is only available from September 2015…"**

*3.6 Discussion and Conclusions*

*L308: Please describe briefly what is new about your method.*

**To briefly clarify what is novel about our method, the following sentence has been included in the first paragraph of the Discussion (Sect. 5): "To our knowledge, this methodological process is the first attempt to quantify synoptic scale vorticity features in sea ice using remote sensing techniques, with the aim to establish an indicator of rotational drift in the sea ice field by which to detect current and future changes in the ice dynamics"**

*L311: Would be more consistent to write that you use four products.*

**We have corrected this, and the sentence now reads: "Four products were used in this study…"**

*L317f: From this formulation, it sounds like the 450 km would be 6-7 times the scale of atmospheric weather. I suggest to reshuffle the sentence between the dashes and write " which is the scale of atmospheric weather and about 6 - 7 times. . . " to be clearer.*

**We have rephrased this sentence as not to incorrectly imply that our choice of radius is 6-7 times larger than atmospheric weather. The sentence now reads: "For this reason, our detection algorithm identifies circular ice drift features with a radius of 450 km ± 50 km – which is the scale of atmospheric weather and about 6-7 times the spatial resolution of the products – and quantifies the characteristics of the vorticity field within its circumference."**

*L321ff: Can you speculate on possible physical or methodological reasons for the discrepancy between a) the single-sensor products and b) the single-sensor products and the merged product? Having an*

*idea about this would add much relevance to the study and help it to be even more than a description of an interesting finding.*

**As mentioned earlier, the discrepancy between products is partly caused by the quality of the drift vectors varying between products. The merged product has the best coverage of good quality drift estimates, followed by the AMSR-2, SSMI/S and ASCAT products. It was also noticed that the discrepancy between products is enhanced when the flag quality restriction is relaxed (results not shown but mentioned in text (Sect. 4.2)). In the revised section, there is more focus on the differences in the coverage of good quality drift estimates between products. This includes an added explanation of the flag meanings in the Methodology (Sect. 3); an additional subsection in the Results (Case Studies 1 and 2 in Sect. 4.1.1 and 4.1.2 respectively) where two maps are included (Fig. 1 and 2); and added text in the Discussion.**

**Based on these additions, we have offered our interpretation of the results in the revised version as follows:**

**"These examples underline the different quality of the drift retrieval between products, which ultimately leads to a difference in the coverage of the computed vorticity field and in the detection of features"**

**and**

**"Our analysis of a few study cases gives some hints that the better coverage of the merged product increases the detection of rotational drift in the ice compared to the single-sensor products... The better coverage of the merged product seemingly makes it a good candidate for synoptic scale vorticity analysis. However, we speculate that the large variability introduced by the merging process may also cause an artificial intensification of cyclonic rotation. This is because the more extreme gradients between adjacent drift vectors in a heterogenous drift-field are manifesting into an exaggerated vorticity field. However, in the absence of independent observations that would corroborate our findings, we are unable to fully identify whether this is an artefact or a feature."**

*L341f: It is not entirely clear to me why more variability would artificially intensify cyclonic rotation. Please explain in more detail.*

**Please see the answer to the previous comment, in which we better explained this remark.**

*L343: "observation" → "observations"*

**This correction has been made in the revised version (Sect. 5).**

*L352: See comment to L279.*

This sentence has been corrected and now reads: "The AMSR-2 detected the same uniformity since its derived drift product became available in September 2015."

*L357-359: Are all the references needed? if not, please pick the two or three most relevant ones.*

We have removed two of these references, and the sentence now reads: "Among other causes that involve atmospheric and oceanic components (Blanchard-Wrigglesworth et al., 2021; Meehl et al., 2019; Stuecker et al., 2017; Wang et al., 2019a), it has been argued that an intensification in polar storms in 2016 contributed to the anomalously quickened SIE decline from 2015…"

*General: It would add much value if you could give the reader a recommendation in which cases the merged product is more appropriate and in which cases one might be better off with one of the single-sensor products.*

While we are reluctant to recommend which product is best suitable for vorticity analysis due to the lack of dedicated experimental data validating the various products at the synoptic scale, we have added the text described a few comments above in the revised discussion. We suggest, based on the visual analysis of selected cases exemplified in the presented case studies 1 and 2, that the better coverage of the merged product likely makes it the best candidate. We do, however, remind the reader that it is not known if the merged product may be amplifying the cyclonic signal detected, or whether the higher cyclonic intensities detected are a good estimate of true conditions. We have included the following text in the Discussion (Sect. 5): "The better coverage of the merged product seemingly makes it a good candidate for synoptic scale vorticity analysis; however, we speculate that the large variability introduced by the merging process may also cause an artificial intensification of cyclonic rotation. This is because the more extreme gradients between adjacent drift vectors in a heterogenous drift-field are manifesting into an exaggerated vorticity field. However, in the absence of independent observations that would corroborate our findings, we are unable to fully identify whether this is an artefact or a feature."

3.7 Code

*Thank you for including the code for the analysis. It would be good, however, if you could add a small readme file to the repository which instructs the user how to use the code. Even though the code is plain and well-written, I think that it would be of value if you could add some comments, so that potential users can use it right away.*

We thank the reviewer for this recommendation. We have taken their advice and updated the online GitHub repository with a README file. The link has been updated in the revised manuscript: https://github.com/waynedejagerUCT/Rotational_Drift_From_Space.git

**Response to Reviewer 2**

Dear Reviewer 2

I would like to thank you for the time you have given to provide us with constructive suggestions to improve the quality of our paper for this second round of review. Your efforts are very much appreciated.

In this document we have provided an overview of the major changes we have made for our revised manuscript after acting upon the suggestions made by you and Reviewer 1. We have also responded to each of your specific comments, and those of Reviewer 1, in a point-by-point format. Your original comment appears in *italics*, while our response appears in **bold**.

*This paper presents an analysis of sea ice vorticity in the Atlantic sector of the Southern Ocean. In particular, it examines 4 datasets of sea ice velocity determined from feature tracking. Three of the datasets are from single sensors, while the fourth is a merged product.*

*The main conclusions seem to be that i) cyclonic ice drift features are more common, intense, and variable than anticyclonic features in this region; ii) the merged product contains more intense cyclonic features than the other products; iii) an increase in cyclonic features occurred after the well-known decline in ice extent in this region around 2014.*

*I believe these results are interesting and important and do deserve publication. However, I have a few issues with the paper as it stands. I was not shown any evidence that the metric used here is an essential climate variable (cyclonic ice drift driven by atmospheric forcing). I also feel that each of the three conclusions I have enumerated above are shown but not explained. So this left me as a reader both confused about exactly what is shown and also wondering why those features have occurred. I think my main problem stems from the fact that all of the figures shown are purely statistical, and do not invoke the physics of sea ice in this region.*

**This manuscript was originally meant to be a brief communication that has been expanded upon request of the handling editor and thanks to the comments of two other reviewers. We agree with the Reviewer that this expanded version does require a proper contextualization of the physical features, which we did not include adequately in the submitted version. We have realized the importance of showing maps of the vorticity features, which would clarify their scales and how they are connected to the atmospheric drivers. As mentioned in the overview, a major change to the manuscript is the addition of another Results subsection (Sect. 4.1), where two case studies are presented. The purpose of these case studies is to help invoke a better understanding of the patterns of movement in the sea-ice field. The two new figures presented here help illustrate the rotational patterns of the sea ice relative to the overlying mean sea level pressure contours, where it can be seen that the vorticity feature is consistent with the structure of the weather feature.**

**Regarding the enumerated conclusions mentioned, we have expanded our discussion to provide possible reasons for these conclusions:**

**"i) cyclonic ice drift features are more common, intense, and variable than anticyclonic features in this region"**

In our revised version, we argue that this "is primarily because atmospheric cyclones are more effective in engendering rotational motion into the underlying sea ice, or whether the feature tracking method of drift retrieval is overly sensitive to the conditions under an atmospheric cyclone." Our work presents the methodology which has led to this information. The underlying mechanisms will be the subject of future work, but we suggest that an improved understanding of how well satellite drift products are able to capture vorticity in sea ice is still needed.

*"ii) the merged product contains more intense cyclonic features than the other products"*

The merged product detected more features (both cyclonic and anticyclonic) primarily due to its substantially better coverage than the single-sensor products. This is now presented with the help of the case studies in the new Sec. 41, where the difference between the products is made evident. Upon request from the other reviewer, we have also improved the new Fig. 3 and 4 (formerly 1 and 2) to show the different distribution of the detected feature. We have included the following text into the revised Discussion: "The better coverage of good quality-flagged drift estimates from the AMSR-2 product means that its resultant vorticity field has better coverage than the SSMI/S product, despite both being passive-microwave based, while the coverage of the active-microwave based ASCAT product is considerably worse than the other three products."

*iii) an increase in cyclonic features occurred after the well-known decline in ice extent in this region around 2014."*

In our original manuscript Discussion (Sect. 5), we provide a possible explanation of this result: "Among other causes that involve atmospheric and oceanic components (Blanchard-Wrigglesworth et al., 2021; Meehl et al., 2019; Stuecker et al., 2017; Wang et al., 2019a), it has been argued that an intensification in polar storms in 2016 contributed to the anomalously quickened SIE decline from 2015, as the overlying winds of these synoptic features induced changes in the sea-ice dynamics (Wang et al., 2019b). Our results show that sea ice in the Atlantic sector was more susceptible to cyclonic rotational features after 2015, which can be interpreted as a response to an increased incidence of polar cyclones. If the sea ice was thinner and more prone to free-drift motion in general, then we would expect both an increase in cyclonic and anticyclonic rotation, but instead, only an increase in the intensity of cyclonic features is detected." We have then expanded our discussion to include that it is necessary to have a dedicated observational effort to assess the qualities of the products and their use to compute vorticity: "Prior to performing further analysis of drift variability and longer-term trends linked to polar atmosphere variability, further validation of the vorticity metric with *in situ* experiments is required to better discern the differences between products. We therefore argue for the need of a concerted experiment to increase the number of observations of sea-ice drift in Antarctic sea ice to assess the quality of the products and their use to quantify rotational features. A better understanding of these features will enable us to confidently use rotational drift of sea ice as a potential derived index to detect climatic trends." We have also modified the abstract to reflect the changes in the revised text and commented on the need for a longer time series of data before any climatic trends can be assessed.

Lastly, We have removed the references to any new essential climate variable and indicated that the proposed measure of rotational drift is a potential index to detect climatic trends in Antarctic sea ice

*As I understand it, the key metric that is being plotted is the grid-scale vorticity averaged over overlapping circles with a 450km radius. The paper frequently refers to atmospheric cyclones/anticyclones and seems to assume throughout that all vorticity detected by this metric is associated with cyclones/anticyclones. However, I don't see that demonstrated anywhere. I can imagine many other things that would create vorticity in these data, e.g. coastal currents, ice edge currents, ocean eddies, tidal effects, artefacts in the data from missing data, interpolation or edge effects from different satellite swaths, etc. One piece of evidence that non-atmospheric-cyclone vorticity is important is that the standard deviation of vorticity within the circles is large. Does that mean there are strong sub-circle features, which means smaller scale, which means not cyclones? So, what is the basis for claiming that the vorticity captured by this metric is from atmospheric cyclones/anticyclones? I think this may be particularly important when just considering the 95th percentile. Perhaps those are the relatively few circles that have strong sub-circle features, e.g. ocean eddies or floes at the ice edge or satellite processing artefacts? I think that demonstrating where the vorticity comes from in a general sense is crucial to the conclusions, and I feel it is missing in the current manuscript.*

**The aim of this paper was not to systematically relate ice-drift rotation with atmospheric features, but to analyze whether existing ice-drift product are sufficiently good to allow such an analysis. Nevertheless, there is evidence in the literature of a tight relationship between synoptic weather and sea-ice movement in the Atlantic sector of the Southern Ocean. We have streamlined the introduction (as also requested by another reviewer) to better highlight our founding argument. Rotational features found in the sea ice may originate from both oceanic and atmospheric drivers, and while some initial studies may indicate that sub-mesoscale oceanic processes under the ice may be concurrent drivers (Biddle and Swart, 2020; Stössel et al., 2018), there is larger evidence of the role played by atmospheric cyclones in driving sea-ice motion. The two cases presented in the newly added Sect. 4.1 present examples of both an atmospheric cyclone and anticyclone, where the scale and location of the sea-ice vorticity features are consistent with the overlying weather conditions. These two cases are examples of other events that can be visually observed when combining the sea-ice drift and atmospheric reanalysis. These particular cases have been selected based on known synoptic events observed during the winter expeditions of the SA Agulhas 2 in the region**

*I think the paper needs to show some maps of vorticity. At present I don't have a good feel for what area is under consideration, or what its vorticity looks like. I haven't studied vorticity plots elsewhere in the literature and so I have no intuition here – maybe I missed something? I think the reader needs to know what the metric is, and I don't think that can be shown by starting with considering higher statistics. I think maps could be constructed such as mean vorticity, mean cyclonic-only vorticity, snapshots of interesting cases (e.g. high cyclonic vorticity), etc.*

**We thank the reviewer for this suggestion, which made us realize the importance of showing individual cases of the vorticity field instead of just the systematic statistical analysis. As mentioned in the answer to the first comment, we included a new subsection in the Results, now Sect. 4.1. This includes two case studies, both of which provide a map of the sea-ice vorticity field and the overlying mean sea level pressure conditions. The purpose of this subsection is to help the reader better visualize the structure of the vorticity features being detected by our algorithm described in the methodology (Sect. 3). Furthermore, the mean intensity, variability and uncertainty of these**

**particular features are also indicated with markers in the modified Fig. 3, 4 and 6. This allows the reader to gain some intuition of what each of these statistical figures represent in the sea-ice drift data. The newly added figure 1 is shown below as a demonstration:**

[Figure]

**Here, for the merged product in panel (d) it is illustrated that the sea-ice velocity field shows an area of negative vorticity (blue) in the same location as the high-pressure cell described in the revised Results Sect. 4.1.2. This map allows the reader to see the relationship between the ice motion and local atmospheric structure. Furthermore, these images also illustrate the discrepancy between products (where each product is a different panel), as well as the area of rejection drift vectors indicated with the dotted hatching.**

*Figure 1 etc. Why are the cyclonic features more intense on average than the anticyclonic features? I think the explanation here really depends upon whether they are dictated by atmospheric forcing or other features. I feel that maps are particularly needed here, to show where the cyclonic/anticyclonic features are and what they look like. If the explanation lies in the atmospheric forcing (cyclones are more intense than anticyclones) then that could be shown and discussed and literature cited.*

**We agree with the reviewer that this feature might be linked to the dominance of extratropical cyclones that establish the background winter weather conditions to which sea-ice dynamics is subjected. This hypothesis is however dependent on the reliability of the satellite products to detect sea-ice drift in a consistent way. In this work we aim to explain the rationale for the detection algorithm and to intercompare the different products. Despite the differences between the single sensors and merged products, there is a consistent indication that anticyclones are less intense than cyclones and there is a change in the intensity of the latter after 2015. We argue for the need of a concerted experiment to increase the number of observations of ice drift in Antarctic sea ice, but the presented results have given us some confidence on the methodology and its eventual application to the study of the underlying drivers. We however prefer to leave this component of the analysis to a further work.**

*Line 225: The merged product has a higher frequency of high cyclonic anomalies than the single-sensor products. Why is this? It is sensitive to the quality flag, but reducing the threshold for that flag just includes data that are suspect and so that is not a good test, in my opinion. Is it because there are more data, e.g. near the ice edge? Does mapping the vorticity fields for the different sensors help here? Mapping the data availability?*

**The newly added Case Studies 1 and 2 (Sect. 4.1) give a better interpretation of how influential the quality flag of the drift estimates are to the resultant vorticity field. This is illustrated with the new Fig. 1 and 2, which provide examples of both atmospheric cyclonic and anticyclonic conditions respectively. Here we show that the vorticity field of the merged product was considerably larger than that of the single-sensor products. In our discussion, we comment: "Unsurprisingly, the merged product has the best coverage in the case of cyclones and anticyclones. This is because the merged product processes drift estimates from multiple sensors and is therefore more likely to have a good quality-flagged drift estimate at each grid point. The better coverage of good quality-flagged drift estimates from the AMSR-2 product means that its resultant vorticity field has better coverage than the SSMI/S product, despite both being passive-microwave based, while the coverage of the active-microwave based ASCAT product is considerably worse than the other three products." Furthermore, it is also shown in Case Study 1 that polar storms tend to modify the ice properties rapidly, reducing the quality of the drift estimates in the affected areas. This often results in single-sensor products being unable to detect cyclonic features in this area, as they fail to meet the valid vorticity threshold based on the flag requirements describe in Sect. 3. The merged product is less susceptible to this, and higher quality drift estimates are retrieved in regions of high cyclonic vorticity.**

*None of the regressions plotted in figures 1 and 2 any statistical significance test. P values should be calculated and quoted on the plots, and the text should be altered wherever the trends are not significant.*

**In our revised version, we have modified these figures (now labelled Fig. 3 and 4) from scatterplots to 2-D histograms (as requested by another reviewer), and included the slope, intercept, standard error of the slop and the number of features on each panel. The 2-D histogram allows for better visualization of data around the line of best fit, particularly in areas of the plot where the data points have a high density. The large number of features detected (~800,000) means that the statistics shown on each panel are robust.**

*Line 137: I understand that for a given time, the circles overlap in space. However, I do not understand how they could overlap in time? And how you could allow them to overlap in time but not space?*

**This comment was meant to inform the reader that the algorithm will never extract more than one feature at the exact same location more than once per 48 h dataset. In other words, there are no duplicate features being extracted, although persistent features may be detected over multiple 48 h datasets. We do, however, acknowledge that this portion of text is unclear and unnecessary, and so it**

has been rephased to "Each of these subdomains represent a vorticity feature, which can partially overlap one another in space.

*Equation (2): Is sigma_tr a displacement error, in units of metres?*

The $\sigma_{tr}^2$ displacement error in the OSI-405-c products are in units of kilometres. We have now included additional text in the revised Methodology section with the units of each variable in Eq. 2.

*Optional: The paper does not consider seasons at all. That seems strange to me as I would imagine that the ice vorticity is very different in the summer and winter, both through the ice mechanics and the atmospheric forcing. I do not think it is essential that the authors introduce seasons to the paper, but I would do that personally as I think it would add a lot of insight.*

This is unfortunately a limitation of the satellite drift products. While we do agree that a seasonal consideration would be interesting and useful, the reliability of the drift products decreases in melting conditions or during times of very low sea-ice coverage (such as very early autumn when the ice band around the continent is narrow). This would mean that a substantially higher number of drift estimates would be rejected based on the status flag criteria described in the methods (Sect. 3). This is further exacerbated by the fact that for every 1 pixel with a rejection quality drift estimate, the vorticity coverage could decrease by 1-4 pixels (due to vorticity value at each pixel requiring a drift estimate at its adjacent pixels). We have included two references (Lavergne, 2016; Sumata et al., 2015) which show that the quality of the drift estimates deteriorate outside of the winter months:

[revised manuscript text omitted]

**TC-144-2021: Response to reviewers of Manuscript Version 3**

**Overview of our revised manuscript after minor changes:**

Note: Reviewer 1 suggested that no further changes are necessary.

Dear Reviewer 2

Once again, we would like to extend our thanks to you for the time you have given to provide us with constructive suggestions to improve the quality of our paper after a second round of review. Your efforts are very much appreciated, as we believe the quality of our manuscript has improved thanks to your suggestions.
In this document we have provided an overview of the major changes we have made to our revised manuscript. We have also responded to each of your specific comments in a point-by-point format. Your original comment appears in italics, while our response appears in **bold**.

*L26: driven by atmospheric*
**The final sentence of the abstract has been rephrased to emphasize that the observed sea-ice variability is driven by local atmospheric conditions, and the sentence now reads:**

**"Such information will allow to confirm whether the detected increase in cyclonic vorticity is linked to rapidly changing sea-ice dynamics driven atmospheric changes and establish the measure of rotational sea-ice drift as a potential indicator of weather driven variability in Antarctic sea ice."**

*Figures 1 and 2: These are fantastic! I am very grateful to the authors for adding these, as I now have a very good feeling for the cyclonic and anticyclonic features they are studying. I had one small thought about the description. In the text the authors understandably focus on the strong local cyclones and anticyclones that they illustrate in the coloured circles. However, there is also a much wider field of vorticity shown in these plots outside those circles, and it is also associated with cyclonic and anticyclonic curvature of the isobars, everywhere as far as I can tell. So perhaps the authors could emphasise that it is not just local storms that are under consideration here, but also the general vorticity field. Relatedly, I see no evidence in either of these figures for any such features as oceanic eddies, boundary jets, coastal currents, etc; the isobars really do seem to explain it all. So maybe the authors could highlight that these figures illustrate their interpretation that the recorded vorticity is dominated by the atmosphere (e.g. on line 395).*

**We thank the reviewer for the enthusiastic feedback. We agree that we should elaborate more on the atmosphere – ice vorticity link in the entire rectangular region, and not just underneath the specific atmospheric feature described. In the revised Case Study 1 (Sect. 4.1.1), we comment on the vorticity field over the Weddell Sea, an area which is not influenced by the intense polar cyclone, but rather by relatively weak high-pressure ridge. We have added:**

**"In addition to the area of ice beneath the atmospheric cyclone, the wider vorticity field is also associated with the cyclonic and anticyclonic curvature of the isobars. All four products detect positive vorticity beneath the elongated high-pressure ridge over the Weddell Sea, 7 185 separating**

two smaller regions of negative vorticity at the ice edge (approximately 62° S and 30° W) and continental coastline (approximately 77° S and 15° W), each of which lie below a pressure trough.”

In the revised Case Study 2 (Sect. 4.1.2), we comment on the negative-to-positive vorticity gradient that is seen across the Weddell Sea, and how the overlying parallel isobars lie perpendicular to this vorticity gradient:

“Again, the vorticity fields detected by all four products show a strong correlation with the curvature of the overlying isobars across the entire rectangular region, much like that shown in Case Study 1 (Sect. 4.1.1). This is particularly visible in Fig. 2 along the negative-to-positive vorticity gradient from the western-to-eastern Weddell Sea, where the parallel isobars lie perpendicular to the vorticity gradient, suggesting the atmospheric pressure gradients control the underlying vorticity field. A region of negative vorticity is also detected near the eastern boundary of the rectangular region which mimics the curvature of the overlying low-pressure cell deflecting eastwards.”

The vorticity field's strong correlation with the overlying atmosphere and the lack of evidence suggesting an ocean sourced influence both give confidence that the vorticity field is mostly weather driven. We therefore elaborate on this in the revised (i) Abstract and (ii) Discussion by adding:

(i)     “Such information will allow to confirm whether the detected increase in cyclonic vorticity is linked to rapidly changing sea-ice dynamics driven atmospheric changes and establish the measure of rotational sea-ice drift as a potential indicator of weather driven variability in Antarctic sea ice.”

(ii)     “The case studies presented (Sect. 4.1) indicate that the vorticity field of the sea ice is strongly linked to the cyclonic and anticyclonic curvature of daily isobars, both in cases of extreme weather events and mild atmospheric conditions. There is no apparent evidence of oceanic drivers effecting ice rotation in this region at these spatial and temporal scales, suggesting that the detected vorticity field is dominated by weather at daily – or even sub-daily – timescales. This aligns with existing literature that sea-ice variability in the Atlantic Sector of the Southern Ocean is primarily driven by local atmospheric conditions (Kwok et al., 2017; Matear et al., 2015).”

*General: It has only just occurred to me that of course there are more cyclones than anticylones in this area, as it is dominated by a climatological low pressure (the Weddell Sea Low)! If the ice vorticity is dominated by the atmosphere then of course this necessarily means there is more cyclonic ice vorticity than anticylconic. So perhaps the authors could mention this? What this paper shows really nicely is precisely how that difference is manifested – in intensity of cyclones etc.*

*L459: This explanation seems to miss the most obvious explanation of all: there is more atmospheric cyclonic forcing! It doesn't have to be more effective, or hide the results, there is just more of it! This is a climatological low pressure area, so that implies there will be negative ice vorticity on average.*

Following the reviewer's comment, we have added extra text to our revised Discussion to include comments on the overall cyclonic rotation in the atmosphere in our studied region, and that in cases whereby ice motion is primarily weather drive – like such is the case here – the dominance of negative vorticity in the ice is likely influenced by the dominance of negative vorticity in the overlying atmosphere. We have added the following into the revised Discussion (Sect. 5):

“Furthermore, the Weddell Sea is dominated by a climatological low-pressure cell – termed the Weddell Low – which is the result of the frequent passing of cyclones through this region caused by intense cyclogenesis in the Atlantic Sector (Grieger et al., 2018; Simmonds et al., 2003; Wei and Qin,

2016) or low pressure features crossing the Drake Passage to the east (Gonzalez et al., 2018). Since it has been shown that the ice vorticity field is primarily weather driven, this dominance of cyclonic rotation in the atmosphere likely contributes to the more frequent and intense cyclonic vorticity features detected in the sea-ice.”

We therefore included this consideration as a possible reason for dominance of negative vorticity detected in the ice:

“It is thus difficult to discern whether the dominance of cyclonic rotation in the ice – both in frequency of events and their intensity – is due to 1) the dominance of cyclonic rotation in the overlying atmosphere; 2) atmospheric cyclones being more effective at engendering rotational motion in the underlying sea ice than anticyclones; 3) the feature tracking method of drift retrieval being overly sensitive to the conditions under an atmospheric cyclone; 4) any combination of the aforementioned considerations.”